# Learning with Dual-level Noisy Correspondence for Multi-modal Entity Alignment

**Haobin Li**[1]   **Yijie Lin**[1]   **Peng Hu**[1]   **Mouxing Yang**[1*]   **Xi Peng**[2,3*]

College of Computer Science, Sichuan University, China.[1]
School of Artificial Intelligence, Sichuan University, China.[2]
National Key Laboratory of Fundamental Algorithms and Models for Engineering
Numerical Simulation, Sichuan University, China.[3]
{haobinli.gm, linyijie.gm, penghu.ml, yangmouxing, pengx.gm}@gmail.com

## Abstract

Multi-modal entity alignment (MMEA) aims to identify equivalent entities across heterogeneous multi-modal knowledge graphs (MMKGs), where each entity is described by attributes from various modalities. Existing methods typically assume that both intra-entity and inter-graph correspondences are faultless, which is often violated in real-world MMKGs due to the reliance on expert annotations. In this paper, we reveal and study a highly practical yet under-explored problem in MMEA, termed Dual-level Noisy Correspondence (DNC). DNC refers to misalignments in both intra-entity (entity-attribute) and inter-graph (entity-entity and attribute-attribute) correspondences. To address the DNC problem, we propose a robust MMEA framework termed RULE. RULE first estimates the reliability of both intra-entity and inter-graph correspondences via a dedicated two-fold principle. Leveraging the estimated reliabilities, RULE mitigates the negative impact of intra-entity noise during attribute fusion and prevents overfitting to noisy inter-graph correspondences during inter-graph discrepancy elimination. Beyond the training-time designs, RULE further incorporates a correspondence reasoning module that uncovers the underlying attribute-attribute connection across graphs, guaranteeing more accurate equivalent entity identification. Extensive experiments on five benchmarks verify the effectiveness of our method against DNC compared with seven state-of-the-art methods. Code is available at https://github.com/XLearning-SCU/2026-ICLR-RULE.

## 1 Introduction

Multi-Modal Entity Alignment (Liu et al., 2021; Li et al., 2023) (MMEA) aims to identify equivalent entities across different Multi-modal Knowledge Graphs (MMKGs) (Liu et al., 2019; Zhu et al., 2022), where each entity is associated with attributes of various modalities (*e.g.*, structural triples and images). Due to the heterogeneity of attributes from different modalities and graphs from different sources (*e.g.*, Wikidata (Vrandečić & Krötzsch, 2014) and YAGO (Suchanek et al., 2007)), the key challenge of MMEA is to learn a comprehensive representation for each entity with its respective attributes while eliminating the cross-graph discrepancy. To this end, existing methods usually conduct multi-modal fusion for attributes within the same entity based on the intra-entity correspondences (*i.e.*, entity-attribute pairs), while performing cross-graph alignment by resorting to the inter-graph correspondences (*i.e.*, entity-entity pairs and attribute-attribute pairs).

Despite significant efforts in intra-entity attribute fusion (Chen et al., 2023a; Huang et al., 2024a) and inter-graph discrepancy elimination (Xu et al., 2023; Guo et al., 2021), existing MMEA methods heavily rely on the assumption of faultless intra-entity and inter-graph correspondences. However, as shown in Fig. 1(a), the assumption is daunting and even impossible to satisfy, leading to the Noisy Correspondence (NC) problem at dual levels. On the one hand, as the MMKG construction requires expert knowledge, it is inevitable to wrongly associate some entities with irrelevant attributes, resulting in intra-entity NC. For instance, image of "Elvis Tsui" is incorrectly associated with entity "Jason Momoa" because of the visual resemblance. On the other hand, due to the inherent

---

*Corresponding author.

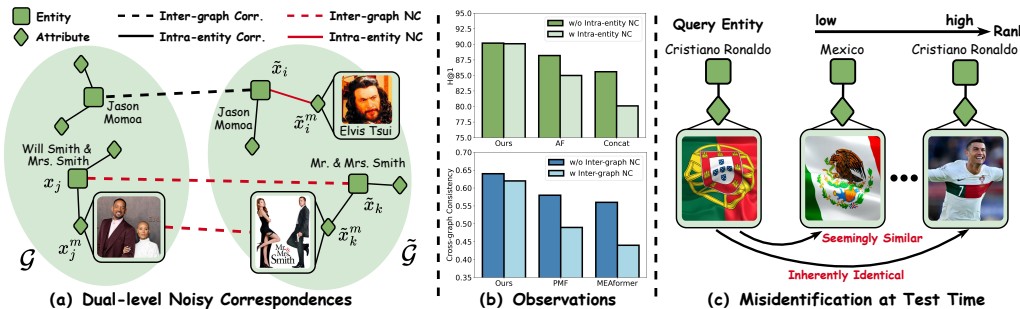

Figure 1: (a) **Dual-level Noisy Correspondence** occurs in both intra-entity level (*i.e.*, entity-attribute pairs such as $(\tilde{x}_i, \tilde{x}_i^m)$) and the inter-graph level (*i.e.*, entity-entity $(x_j, \tilde{x}_k)$ or attribute-attribute pairs $(x_j^m, \tilde{x}_k^m)$). (b) **Observations**: On the one hand, both vanilla adaptive fusion (AF) and concatenation (Concat) tend to integrate erroneous attributes and thus degrade performance, while our method achieves reliable fusion against inter-graph NC. On the other hand, existing methods suffer in cross-graph alignment when encountering inter-graph NC, whereas our method boosts performance by mitigating the negative impact of intra-entity NC. (c) **Misidentification at Test Time**: Seemingly similar attribute pairs may prevent the query entity from being associated with its equivalent entity. For instance, the implicit connection between the football player "Cristiano Ronaldo" and his home country are often overlooked, resulting in misidentification during entity alignment.

complexities in attribute and entity association, accurately associating all the inter-graph entities and their corresponding attributes is impractical, leading to inter-graph NC. For example, movie entity "Mr. & Mrs. Smith" is mistakenly associated with real-life couple "Will Smith and Mrs. Smith". According to the statistics in Appendix B, real-world benchmarks always contain numerous NC (*e.g.*, over 50% in ICEWS benchmarks). As shown in Fig. 1(b), NC would not only undermine the fusion of within-entity attributes but also mislead the inter-graph alignment, both of which significantly degrade the performance.

Based on the above observations, we reveal a new problem for MMEA, termed Dual-level Noisy Correspondence (DNC). To conquer the DNC problem, we propose a novel method, dubbed dually RobUst LEarning (RULE), for achieving robust MMEA against DNC. Specifically, RULE first estimates the reliability of both the intra-entity and inter-graph correspondences by resorting to a dedicatedly-designed two-fold principle and then divides the entity-attribute, entity-entity, and attribute-attribute pairs into different groups. Based on the estimated reliabilities and division results, RULE alleviates the negative impact of intra-entity NC during intra-entity attribute fusion, while preventing the model from overfitting the inter-graph NC during inter-graph discrepancy elimination. Beyond the training-time designs, RULE further incorporates a novel correspondence reasoning module to enhance the test-time robustness. In brief, this module performs deep reasoning to uncover the underlying attribute-attribute connections across graphs, thus preventing seemingly dissimilar but inherently identical attributes from being neglected (as shown in Fig. 1(c)) and guaranteeing more accurate equivalent entity identification during inference.

In summary, the major contributions and novelties of this work are given as follows.

- We reveal and study a novel and practical problem in MMEA, termed Dual-level Noisy Correspondence (DNC). In brief, DNC refers to the noisy correspondence rooted in the intra-entity (entity-attribute) pairs and inter-graph (entity-entity, attribute-attribute) pairs. We empirically demonstrate that DNC not only undermines multi-modal attribute fusion but also misleads the inter-graph alignment, leading to significant performance degradation for existing MMEA methods.

- To achieve robust MMEA against the DNC problem, we propose a novel method termed RULE, which estimates the reliability of both the intra-entity and inter-graph correspondences with a dedicatedly-designed two-fold principle and accordingly mitigates the negative impact of DNC during the multi-modal attribute fusion and inter-graph alignment processes.

- During inference, RULE employs a novel correspondence reasoning module to uncover inherently-identical attributes and accordingly achieve more precise cross-graph equivalent entity identifi-

cation. To the best of our knowledge, this could be one of the first methods to enhance test-time robustness for the MMEA task.

## 2 METHOD

In this section, we introduce the proposed RULE for tackling the DNC problem. In Section 2.1, we present the formal definition of the MMEA task and the DNC problem. In Section 2.2, we elaborate on the two-fold principle for the reliability estimation and pair division. In Section 2.3-2.4, we introduce the robust attribute fusion and robust discrepancy elimination modules. In Section 2.5, we design a test-time correspondence reasoning module to uncover underlying connections between inter-graph attributes, facilitating the equivalent entity identification.

### 2.1 PROBLEM FORMULATION

Given two heterogeneous multi-modal knowledge graphs (MMKGs), denoted as $\mathcal{G} = \{x_i, \{x_i^m\}_{m=1}^M\}_{i=1}^N$ and $\tilde{\mathcal{G}} = \{\tilde{x}_j, \{\tilde{x}_j^m\}_{m=1}^{\tilde{M}}\}_{j=1}^{\tilde{N}}$, where $x_i$ and $\tilde{x}_j$ are entities in $\mathcal{G}$ and $\tilde{\mathcal{G}}$, respectively. Each entity $x_i \in \mathcal{G}$ is associated with $M$ attribute-specific attributes $\{x_i^m\}_{m=1}^M$, such as structured triples, textual descriptions, and images.

Within a single graph, the association between an entity and its attributes is captured by entity-attribute pairs $(x_i, x_i^m, h_i^m)$, where $h_i^m \in \{0, 1\}$ is a binary indicator, $h_i^m = 1$ indicates the valid *intra-entity correspondence*, and $h_i^m = 0$ denotes no correspondence between $x_i$ and $x_i^m$. Across graphs, *inter-graph correspondences* govern the alignment of both the entity-entity pairs and attribute-attribute pairs. To be specific, the entity-entity pair is represented by $(x_i, \tilde{x}_j, y_{ij})$, where the correspondence $y_{ij} = 1$ if $x_i$ and $\tilde{x}_j$ refer to the same real-world concepts, and $y_{ij} = 0$ otherwise. Similarly, the attribute-attribute pair is denoted by $(x_i^m, \tilde{x}_j^m, y_{ij}^m)$, where the correspondence $y_{ij}^m = 1$ *i.f.f* both attributes are linked to correct entities (*i.e.*, $h_i^m = 1$ & $\tilde{h}_j^m = 1$) and the corresponding entities $x_i$ and $\tilde{x}_j$ are aligned (*i.e.*, $y_{ij} = 1$). In other words, once the inter-graph entities are associated, their corresponding attributes could be treated as matched.

Given a query entity $x_i \in \mathcal{G}$, the goal of multi-modal entity alignment is to identify its equivalent entity $\tilde{x}_j$ from the other $\tilde{\mathcal{G}}$ such that $y_{ij} = 1$. To this end, existing approaches typically follow a two-stage pipeline: i) intra-entity attribute fusion: for each entity $x_i$, attribute representations are first extracted using attribute-specific encoders $z_i^m = f^m(x_i^m)$, and then aggregated to form a unified entity representation $z_i$; ii) inter-graph discrepancy elimination: based on the fused entity representations $z_i$ and $\tilde{z}_j$, contrastive learning (Chen et al., 2020) is employed to mitigate the inter-graph discrepancy. However, in practice, this pipeline assumes that both the intra-entity correspondences (*i.e.*, entity-attribute $h_i^m$) and inter-graph correspondences (*i.e.*, entity-entity $y_{ij}$ and attribute-attribute $y_{ij}^m$) are perfectly labeled. However, due to annotation errors, such an assumption is often violated, leading to the DNC challenge. As discussed in Introduction, the DNC problem would undermine the inter-graph and intra-entity learning, leading to remarkable performance degradation.

### 2.2 RELIABILITY ESTIMATION AND PAIR DIVISION

To facilitate robust inter-graph discrepancy elimination and intra-entity attribute fusion, we first estimate the reliability of both the intra-entity and inter-graph correspondences by resorting to a two-fold principle, *i.e.*, uncertainty and consensus. Without loss of generality, in the following, we take the inter-graph entity-entity correspondence as a showcase to elaborate on the process of correspondence reliability estimation. For a given entity $x_i$, the reliability $w_i$ between $x_i$ and its associated counterpart $\tilde{x}_j$ ($y_{ij} = 1$) is estimated using the following principle:

$$w_i = (1 - u_i)\gamma + c_i(1 - \gamma), \tag{1}$$

where $\gamma$ is the balanced hyper-parameters (fixed as $0.5$ for simplicity, see Appendix G.10 for more choices), $u_i$ and $c_i$ denote the uncertainty and consensus for the correspondence and will be detailed in the following sections.

#### 2.2.1 UNCERTAINTY MODELING

For a given entity, uncertainty in this work refers to whether its correspondence is trustworthy or not, which could serve as the principle to identify NC. According to the Dempster-Shafer Theory (Shafer,

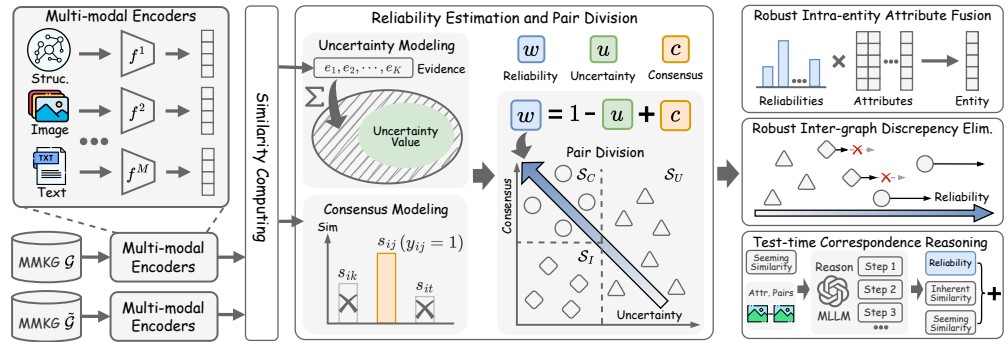

Figure 2: Overview of our method RULE. Given two MMKGs $\mathcal{G}$ and $\tilde{\mathcal{G}}$, RULE first projects the entity attributes into a shared latent space and computes cross-graph attribute similarities. These similarities are used to estimate the reliability of inter-graph correspondences and categorize cross-graph pairs into three subsets: $\mathcal{S}_C$, $\mathcal{S}_I$, and $\mathcal{S}_U$. Subsequently, the robust intra-entity attribute fusion module and robust inter-graph discrepancy elimination are employed to mitigate the impact of both intra-entity and inter-graph noisy correspondences. Beyond training-time robustness, a test-time correspondence reasoning module uncovers latent attribute-attribute connections across graphs, enabling more accurate equivalent entity identification during inference.

1992), uncertainty could be quantified by evidence, which measures how the data support the association between a query and a candidate. Specifically, the more evidences the entity accumulates, the lower uncertainty it embraces. Formally, evidence of the entity pairs $(x_i, \tilde{x}_j)$ is defined as

$$e_{ij} = \exp\left(\tanh\left(s_{ij}/\tau\right)\right), \tag{2}$$

where $s_{ij} = z_i \cdot \tilde{z}_j$ denotes the dot product between the entity representation $z_i$ and $\tilde{z}_j$, $\tau$ is the temperature, and the evidence vector for $x_i$ is $\boldsymbol{e}_i = [e_{i1}; e_{i2}; \cdots; e_{i\tilde{N}}]$. Following Subjective Logic (Sensoy et al., 2018), we associate the evidence vector $\boldsymbol{e}_i$ with the parameters of the Dirichlet distribution $\boldsymbol{\alpha}_i = [\alpha_{i1}, \alpha_{i2}, \cdots, \alpha_{i\tilde{N}}]$, where $\alpha_{ij} = e_{ij} + 1$.

**Definition 1.** *Uncertainty. For a given entity $x_i$, the uncertainty and the corresponding belief mass are defined as*

$$u_i = \frac{\tilde{N}}{Q_i} \text{ and } b_{ij} = \frac{e_{ij}}{Q_i} = \frac{\alpha_{ij} - 1}{Q_i}, \tag{3}$$

*where $Q_i = \sum_j^{\tilde{N}} (e_{ij} + 1) = \sum_j^{\tilde{N}} \alpha_{ij}$ and $u_i + \sum_j^{\tilde{N}} b_{ij} = 1$.*

The $Q_i$ denotes the Dirichlet distribution strength, and the belief mass assignment $\boldsymbol{b}_i = [b_{i1}; b_{i2}; \cdots; b_{i\tilde{N}}]$, *i.e.*, subjective opinion, corresponds to the Dirichlet distribution with parameters $\boldsymbol{\alpha}_i$. Such a formulation encourages the mismatched entity-entity pairs to yield limited evidence, as the given entity fails to associate with any entity in the other MMKG, resulting in high uncertainty.

### 2.2.2 CONSENSUS MODELING

Although the formulated uncertainty would help to identify noisy correspondence, we observe that a low uncertainty does not necessarily indicate a correct correspondence. Formally,

**Theorem 1.** *A low uncertainty $u_i$ does not necessarily imply that the highest belief is assigned to the annotated correspondence $\boldsymbol{y}_i$, i.e.,*

$$z_i \text{ with low } u_i \not\Rightarrow \arg\max \boldsymbol{b}_i = \arg\max \boldsymbol{y}_i. \tag{4}$$

The Proof is placed in Appendix E. Here, $\boldsymbol{y}_i = [y_{i1}; y_{i2}; \cdots; y_{i\tilde{N}}]$ is a one-hot vector indicating the inter-graph entity-entity correspondence of entity $x_i$. Such a theorem highlights that uncertainty is insufficient to determine whether the belief is concentrated on the annotated correspondence. Therefore, we propose the consensus principle as follows.

**Definition 2.** *Consensus. For a given entity $x_i$, the consensus is defined as*

$$c_i = \max(0, \boldsymbol{s}_i \cdot \boldsymbol{y}_i), \tag{5}$$

where $\boldsymbol{s}_i = [s_{i1}, s_{i2}, \cdots, s_{iN}]$ denotes the similarity vector, $\max(0, \cdot)$ ensures the consensus is non-negative.

Intuitively, a low consensus $c_i$ indicates that the given correspondence is unreliable, thus serving as another principle to identify noisy correspondence. However, during inference, the annotated correspondence $\boldsymbol{y}_i$ in Eq. 5 is unavailable. To remedy this, we propose to estimate the correct correspondence through a greedy strategy based on marginal contribution. Here, we begin with a definition of marginal contribution.

**Definition 3.** *For a given entity $x_i$, the marginal contribution of its $m$-th attribute is defined as*

$$\Delta = v(\pi \cup \{m\}) - v(\pi), \tag{6}$$

*where $v(\cdot)$ indicates the value function, $\pi \subseteq \Pi \setminus \{m\}$ denotes a subset $\pi$ of attributes excluding the $m$-th one, $\Pi$ is the complete set of available attributes.*

In the implementation, we define the value function as $v(\pi) = \max\left(\frac{1}{|\pi|} \sum_{j \in \pi} \boldsymbol{s}_i^j\right)$ and $v(\pi \cup \{m\}) = \max\left(\frac{1}{|\pi|+1} \sum_{j \in \pi \cup \{m\}} \boldsymbol{s}_i^j\right)$, where $|\cdot|$ denotes the number of attributes. Inspired by Shannon's principle that "the essence of information is to eliminate uncertainty", we expect that the informative attributes would contribute to establishing reliable correspondence for the entity-attribute pairs. Thus,

**Assumption 1.** *For a given entity $x_i$, if $x_i^m$ is correctly associated with $x_i$, then $\Delta \geq 0$. Conversely, if $x_i^m$ is irrelevant to $x_i$, then $\Delta < 0$.*

Assumption 1 provides a feasible way to estimate the correct correspondence. Specifically, incorporating attributes until the marginal contribution no longer improves, and the established subset $\pi$ would help to indicate a reliable correspondence. To implement this, we adopt the following greedy strategy,

$$\pi^* = \pi_0 \cup \{m \in (\Pi \setminus \pi_0) \mid v(\pi_0 \cup \{m\}) - v(\pi_0) > 0\}, \tag{7}$$

where $\pi_0$ denotes the initial subset with $|\pi_0| = \lfloor \frac{M}{2} + 1 \rfloor$ when $M \geq 3$. See more details in Appendix F.3. With the selected subset $\pi^*$, the estimated correspondence is finally given as $\boldsymbol{y}_i = \text{one-hot}(\arg\max(\frac{1}{|\pi^*|} \sum_{m \in \pi^*} \boldsymbol{s}_i^m))$, where one-hot denotes the vector conversion.

### 2.2.3 PAIR DIVISION

With the formulated uncertainty and consensus, we could further identify the inter-graph NC. Specifically, we propose to divide the inter-graph pairs with $y_{ij} = 1$ into three portions: noisy portion with high uncertainty $\mathcal{S}_U = \{x_i, \tilde{x}_j \mid u_i > \beta_u\}$, noisy portion with low consensus $\mathcal{S}_I = \{x_i, \tilde{x}_j \mid u_i \leq \beta_u \text{ and } c_i < \beta_c\}$ and clean portion $\mathcal{S}_C = \{x_i, \tilde{x}_j \mid u_i \leq \beta_u \text{ and } c_i \geq \beta_c\}$. The thresholds $\beta_u$ and $\beta_c$ are determined in a self-adaptive manner via

$$\beta_u = \min(u^{TP}, 1 - \beta), \quad \beta_c = \max(\beta, c^{TP}), \tag{8}$$

where $u^{TP} = \max_{i \in \mathcal{S}^{TP}} u_i$, $c^{TP} = \min_{i \in \mathcal{S}^{TP}} c_i$, and $\beta$ indicates the threshold hyperparameter. Here, $\mathcal{S}^{TP} = \{i \mid \arg\max(\boldsymbol{s}_i) = \arg\max(\boldsymbol{y}_i)\}$ denotes the set of true positive pairs. With the above pair division, the inter-graph pairs could be divided into $\mathcal{S}_U$, $\mathcal{S}_I$, and $\mathcal{S}_C$, which are further used for inter-graph discrepancy elimination.

### 2.3 ROBUST INTER-GRAPH DISCREPANCY ELIMINATION

With the established reliability and pair division results, we could obtain three subsets: $\mathcal{S}_U$, $\mathcal{S}_I$, and $\mathcal{S}_C$. Since the pairs in $\mathcal{S}_U$ exhibit high uncertainty, they are considered as unreliable and are excluded from the discrepancy elimination. As discussed in Section 2.2.2, inter-graph pairs with low consensus do not necessarily indicate correct matches, thus the pairs in $\mathcal{S}_I$ cannot be regarded as reliable. Accordingly, we propose a novel Dually Robust Learning (DRL) that employs tailored strategies for the three subsets, thereby achieving robustness against inter-graph noisy correspondence. Formally, the overall objective is defined as

$$\mathcal{L} = \mathcal{L}_{DR} + \lambda \mathcal{L}_{Reg}, \tag{9}$$

where $\mathcal{L}_{DR}$ and $\mathcal{L}_{Reg}$ denotes the dually robust loss and regularization loss, $\lambda$ denotes the trade-off parameter. Specifically, the dually robust loss and regularization loss are given by,

$$\mathcal{L}_{DR} = \mathcal{L}_{DR}(\boldsymbol{\alpha}_i, \hat{\boldsymbol{y}}_i) + \sum_{m=1}^{M} \mathcal{L}_{DR}(\boldsymbol{\alpha}_i^m, \hat{\boldsymbol{y}}_i^m), \quad \mathcal{L}_{Reg} = \mathcal{L}_{Reg}(\boldsymbol{\alpha}_i, \hat{\boldsymbol{y}}_i) + \sum_{m=1}^{M} \mathcal{L}_{Reg}(\boldsymbol{\alpha}_i^m, \hat{\boldsymbol{y}}_i^m), \quad (10)$$

where $\boldsymbol{\alpha}_i^m$ and $\hat{\boldsymbol{y}}_i^m$ are the Dirichlet parameter and refined correspondence for $x_i^m$. More specifically, for the given entity $x_i$, the dually robust loss is defined as

$$\mathcal{L}_{DR}(\boldsymbol{\alpha}_i, \hat{\boldsymbol{y}}_i) = \mathbb{I}(i \notin \mathcal{S}_U) \int \|\hat{\boldsymbol{y}}_i - \boldsymbol{p}_i\|_2^2 \, D(\boldsymbol{p}_i \mid \boldsymbol{\alpha}_i) \, d\mathbf{p}_i, \quad (11)$$

where $D(\boldsymbol{p}_i \mid \boldsymbol{\alpha}_i)$ denotes the density function of the Dirichlet distribution over the query probability $\boldsymbol{p}_i = [p_{i1}, p_{i2}, \cdots, p_{i\tilde{N}}]$, and $\mathbb{I}(\cdot)$ indicates an indicator function evaluating to 1 *i.f.f* the condition is satisfied. The refined correspondence $\hat{\boldsymbol{y}}_i$ is defined as follows,

$$\hat{\boldsymbol{y}}_i = \begin{cases} \boldsymbol{y}_i, & \text{if } i \in \mathcal{S}_C \\ c_i \boldsymbol{y}_i + (1 - c_i) \operatorname{Softmax}(\boldsymbol{s}_i), & \text{if } i \in \mathcal{S}_I \end{cases}. \quad (12)$$

Such behavior enhances robustness against inter-graph noisy correspondences for the following reasons. On the one hand, the upper bound of query probability is proportional to $Q_i$ (Theorem 2), thus preventing over-optimization when the accumulate $Q_i$ is limited. On the other hand, excluding high-uncertainty correspondences in $\mathcal{S}_U$ and refining the low-consensus correspondences in $\mathcal{S}_I$ would prevent erroneous optimization caused by NC.

Although the proposed dually robust loss in Eq. 11 could encourage higher evidence for inter-graph pairs with reliable correspondence, it is unable to guarantee that unassociated inter-graph pairs generate limited evidence. To achieve this, a Kullback-Leibler (KL) divergence term is adopted to penalize the evidence of the unassociated inter-graph pairs, *i.e.*,

$$\mathcal{L}_{\text{Reg}}(\boldsymbol{\alpha}_i, \hat{\boldsymbol{y}}_i) = \text{KL}\left[D\left(\boldsymbol{p}_i \mid \tilde{\boldsymbol{\alpha}}_i\right) \| D\left(\boldsymbol{p}_i \mid \mathbf{1}\right)\right] \quad (13)$$

where $\mathbf{1} \in \mathbb{R}^{\tilde{N}}$ is a $\tilde{N}$-dimensional vector of ones, $\tilde{\boldsymbol{\alpha}}_i = \hat{\boldsymbol{y}}_i + (1 - \hat{\boldsymbol{y}}_i) \odot \boldsymbol{\alpha}_i$ denotes the Dirichlet parameters which help to penalize the evidence of unassociated correspondence, $\Gamma(\cdot)$ and $\psi(\cdot)$ are the gamma and digamma function, respectively.

## 2.4 Robust Intra-entity Attribute Fusion

As discussed in Section 2.1, inter-graph attribute associations emerge as the by-product of establishing entity-attribute and entity-entity correspondences. Therefore, for correctly paired entities, the attribute-attribute correspondence is incorrect, *i.f.f*, the corresponding entity-attribute correspondence is wrongly established. Thus, the inter-graph reliability $w_i^m$ could be employed to identify unreliable intra-entity attributes and weaken the emphasis on them during attribute fusion. Specifically, for a given entity $x_i$, we employ the following Dually Robust Fusion (DRF) module to obtain the integrated representation,

$$z_i = \oplus_{m \in M} \left(w_i^m \cdot z_i^m\right), \quad (14)$$

where $\oplus$ indicates the concatenation operator. Such behavior achieves robustness against noisy entity-attribute pairs by fusing the multi-modal attributes with adaptive weights. In other words, attributes with higher reliability are emphasized, while those with lower reliability are weakened.

## 2.5 Test-time Correspondence Reasoning

As discussed in the Introduction, the seemingly similar attributes might hinder the identification of equivalent entities. To solve the problem, we propose Test-time correspondence Reasoning (TTR) module, which uncovers the underlying attribute-attribute connections across graphs, thus improving the equivalent entity identification during inference. Specifically, the refined entity-entity similarity scores are given by,

$$\hat{\boldsymbol{s}}_i = \sum_{m \in M} \hat{w}_i^m \cdot \hat{\boldsymbol{s}}_i^m, \quad (15)$$

where $\hat{\boldsymbol{s}}_i^m$ represents the similarity scores of the $m$-th attribute output by the MLLM and $\hat{w}_i^m$ denotes the corresponding reliability weight. Such behavior could mitigate the negative impact of intra-entity

Table 1: Comparisons with state-of-the-art methods on Non-name benchmarks under DNC setting. "Inherent DNC" refers to the setting without any additional injected noise. H@$k$ indicates the top-$k$ retrieval accuracy while MRR denotes the mean reciprocal rank. The best and second best results are marked in **bold** and underline.

| Setting | Method | ICEWS-WIKI | | | ICEWS-YAGO | | | DBP15K ZH-EN | | | DBP15K JA-EN | | | DBP15K FR-EN | | | Avg. |
|---|---|---|---|---|---|---|---|---|---|---|---|---|---|---|---|---|---|
| | | H@1 | H@5 | MRR | H@1 | H@5 | MRR | H@1 | H@5 | MRR | H@1 | H@5 | MRR | H@1 | H@5 | MRR | H@1 |
| Inherent DNC | EVA | 29.6 | 40.7 | 35.1 | 8.0 | 13.7 | 11.1 | 70.7 | 86.8 | 77.9 | 73.6 | 89.5 | 80.6 | 74.3 | 90.5 | 81.4 | 51.2 |
| | MCLEA | 43.2 | 63.1 | 52.4 | 30.1 | 47.7 | 38.8 | 76.6 | 90.8 | 83.0 | 77.8 | 92.0 | 84.1 | 78.7 | 92.7 | 84.9 | 61.3 |
| | XGEA | 49.8 | 61.5 | 55.5 | 35.5 | 46.7 | 41.2 | 81.1 | 93.0 | 86.3 | 82.6 | 94.3 | 87.8 | 83.1 | 94.7 | 88.3 | 66.4 |
| | MEAformer | 53.5 | 70.1 | 61.3 | 35.0 | 51.2 | 42.8 | 82.4 | 93.5 | 87.3 | 81.9 | 94.2 | 87.3 | 82.1 | 94.4 | 87.5 | 67.0 |
| | UMAEA | 51.2 | 70.0 | 59.9 | 32.4 | 49.4 | 40.6 | 79.1 | 93.2 | 85.3 | 79.6 | 93.9 | 85.8 | 81.2 | 95.0 | 87.3 | 64.7 |
| | PMF | 52.6 | 67.9 | 59.9 | 38.3 | 53.2 | 45.4 | 83.9 | 94.6 | 88.9 | 83.9 | 94.9 | 89.0 | 84.4 | 95.3 | 89.6 | 68.6 |
| | HHEA | 49.0 | 64.6 | 56.4 | 37.5 | 50.4 | 43.8 | 48.7 | 62.5 | 55.5 | 49.9 | 60.6 | 55.4 | 52.8 | 63.6 | 58.2 | 47.6 |
| | Ours | 64.2 | 76.7 | 70.0 | 48.8 | 60.5 | 54.6 | 85.6 | 94.8 | 89.7 | 85.2 | 95.4 | 89.6 | 85.1 | 95.4 | 89.6 | 73.8 |
| 20% DNC | EVA | 15.2 | 21.6 | 18.4 | 0.2 | 0.4 | 0.4 | 51.0 | 70.2 | 59.7 | 54.5 | 73.4 | 63.1 | 53.4 | 73.8 | 62.6 | 34.9 |
| | MCLEA | 34.5 | 53.6 | 43.5 | 24.6 | 40.4 | 32.5 | 69.9 | 85.7 | 77.0 | 70.1 | 85.6 | 77.2 | 70.7 | 87.3 | 78.1 | 54.0 |
| | XGEA | 40.4 | 48.4 | 44.6 | 22.6 | 27.6 | 25.7 | 76.3 | 90.7 | 82.7 | 76.6 | 91.1 | 83.0 | 76.9 | 91.2 | 83.7 | 58.6 |
| | MEAformer | 50.8 | 67.5 | 58.4 | 35.9 | 50.7 | 43.0 | 77.7 | 90.6 | 83.4 | 77.8 | 90.9 | 83.6 | 78.0 | 91.5 | 84.0 | 64.0 |
| | UMAEA | 48.4 | 64.6 | 56.1 | 31.1 | 46.5 | 38.6 | 74.5 | 89.6 | 81.3 | 73.6 | 89.4 | 80.7 | 74.3 | 89.9 | 81.1 | 60.4 |
| | PMF | 45.4 | 60.6 | 52.6 | 36.2 | 49.9 | 42.7 | 76.7 | 90.2 | 82.7 | 76.5 | 89.9 | 82.5 | 77.1 | 90.7 | 83.2 | 62.4 |
| | HHEA | 47.8 | 61.8 | 54.4 | 37.4 | 49.5 | 43.3 | 48.7 | 58.8 | 53.8 | 49.0 | 58.7 | 54.0 | 52.5 | 61.7 | 57.1 | 47.1 |
| | Ours | 62.4 | 75.1 | 68.5 | 48.3 | 59.5 | 53.9 | 81.1 | 92.0 | 86.0 | 80.5 | 92.2 | 85.6 | 80.5 | 92.2 | 85.8 | 70.6 |
| 50% DNC | EVA | 0.5 | 0.8 | 0.9 | 0.0 | 0.1 | 0.2 | 17.2 | 30.5 | 23.6 | 18.3 | 32.0 | 24.8 | 14.0 | 27.2 | 20.3 | 10.0 |
| | MCLEA | 24.5 | 39.9 | 31.9 | 17.4 | 31.1 | 24.1 | 55.2 | 72.1 | 63.1 | 54.0 | 70.4 | 61.5 | 54.6 | 70.9 | 62.0 | 41.1 |
| | XGEA | 39.5 | 47.0 | 43.4 | 23.7 | 27.8 | 26.3 | 67.9 | 83.6 | 74.9 | 68.0 | 83.8 | 75.0 | 68.0 | 83.9 | 75.1 | 53.4 |
| | MEAformer | 42.4 | 58.8 | 50.1 | 30.6 | 45.0 | 37.5 | 68.1 | 83.7 | 75.1 | 62.9 | 80.3 | 70.8 | 65.8 | 82.6 | 73.4 | 54.0 |
| | UMAEA | 37.8 | 55.0 | 46.0 | 25.4 | 40.0 | 32.5 | 64.8 | 82.1 | 72.7 | 58.1 | 78.5 | 67.2 | 61.8 | 80.9 | 70.3 | 49.6 |
| | PMF | 35.1 | 48.8 | 41.8 | 29.6 | 42.4 | 35.8 | 67.1 | 82.6 | 74.2 | 65.6 | 80.7 | 72.5 | 66.1 | 81.5 | 73.1 | 52.7 |
| | HHEA | 43.9 | 57.7 | 50.4 | 34.3 | 46.2 | 40.2 | 45.5 | 55.2 | 50.3 | 46.4 | 55.4 | 51.2 | 50.1 | 59.1 | 54.7 | 44.1 |
| | Ours | 58.2 | 69.7 | 63.6 | 46.9 | 57.4 | 52.0 | 73.4 | 85.9 | 79.2 | 71.8 | 84.9 | 77.8 | 71.4 | 84.8 | 77.5 | 64.3 |

NC, which might undermine attribute fusion during test time. More specifically, we employ Chain-of-Thought (Wei et al., 2022; Li et al., 2026) (CoT) to guide the MLLM toward step-by-step reasoning. Mathematically,

$$\hat{\boldsymbol{s}}_i^m = \text{Softmax}\left(\oplus_{j \in \mathcal{T}_i^m}\left(\text{CoT}\left[x_i^m, \tilde{x}_j^m, \boldsymbol{s}_i^m\right]\right)\right), \tag{16}$$

where $\mathcal{T}_i^m$ denotes the set of correspondences with the highest similarity in prior results $\boldsymbol{s}_i^m$, CoT indicates the reasoning process. Although a feasible solution is to prompt the MLLM with simple instructions such as "Identify the similarities between these attributes.", such vanilla prompts fail to fully activate the deep reasoning capabilities of MLLM. In contrast, the proposed CoT-based reasoning would enable the MLLM to leverage prior results and detailed steps for reasoning, preventing deviations from the prior knowledge while facilitating the mining of underlying connections. See Appendix F.5 and Appendix I for more details. Finally, the joint similarity score could be derived as $\boldsymbol{s}_i^{joint} = \boldsymbol{s}_i + \hat{\boldsymbol{s}}_i$ and the identified equivalent entity is given by $\arg\max \boldsymbol{s}_i^{joint}$.

## 3 EXPERIMENTS

In this section, we conduct extensive experiments on five widely-used MMEA datasets to validate the effectiveness of the proposed RULE. Due to space limitation, we present more experiments in Appendix G.

### 3.1 IMPLEMENTATION DETAILS AND EXPERIMENTAL SETTINGS

Our method contains two networks, the attribute-specific encoders $f^m$ and the test-time correspondence reasoning module. Specifically, we first utilize a pre-trained CLIP model (Radford et al., 2021) to extract features from visual and textual attributes. After that, we employ the attribute-specific encoders to obtain the latent embeddings following (Huang et al., 2024a; Xu et al., 2023). For the test-time correspondence module, we use Qwen2.5-VL-72B-Instruct (Bai et al., 2025) as default to facilitate the test-time correspondence reasoning module (Section 2.5). Regarding hyperparameters, we set the trade-off parameter $\lambda$ in Eq. 9, the threshold $\beta$ in Eq. 8 are fixed as $1e^{-4}$, $0.3$ for all the experiments, respectively. The temperature $\tau$ in Eq. 2 is set to $0.07$ following (Chen et al., 2020).

We evaluate our method on five benchmark datasets: ICEWS-WIKI (Jiang et al., 2024), ICEWS-YAGO, DBP15K$_{ZH-EN}$ (Liu et al., 2021), DBP15K$_{JA-EN}$, and DBP15K$_{FR-EN}$. Details of the dataset and evaluation metric are provided in Appendix F.1 and F.4. As discuss in Introduction, the MMEA benchmarks including ICEWS are always contaminated by DNC which denoted as "Inherent DNC"

Table 2: Comparisons with state-of-the-art methods on All-attributes benchmarks under DNC setting.

| Setting | Method | ICEWS-WIKI | | | ICEWS-YAGO | | | DBP15K_ZH-EN | | | DBP15K_JA-EN | | | DBP15K_FR-EN | | | Avg. |
|---|---|---|---|---|---|---|---|---|---|---|---|---|---|---|---|---|---|
| | | H@1 | H@5 | MRR | H@1 | H@5 | MRR | H@1 | H@5 | MRR | H@1 | H@5 | MRR | H@1 | H@5 | MRR | H@1 |
| Inherent DNC | EVA | 90.7 | 95.7 | 93.0 | 86.5 | 94.0 | 89.8 | 89.8 | 96.6 | 92.8 | 94.8 | 98.8 | 96.5 | 98.7 | 99.8 | 99.2 | 92.1 |
| | MCLEA | 93.8 | 98.3 | 95.9 | 92.1 | 97.7 | 94.6 | 94.5 | 98.6 | 96.4 | 97.8 | 99.7 | 98.7 | 99.2 | 99.9 | 99.5 | 95.5 |
| | XGEA | 83.5 | 94.4 | 88.6 | 93.9 | 97.3 | 95.8 | 91.4 | 97.4 | 94.1 | 94.3 | 98.0 | 96.0 | 97.3 | 99.3 | 98.2 | 92.1 |
| | MEAformer | 95.9 | 98.8 | 97.2 | 93.8 | 97.9 | 95.7 | 96.7 | 99.0 | 97.7 | 98.8 | 99.8 | 99.3 | 99.6 | 100.0 | 99.8 | 97.0 |
| | UMAEA | 94.8 | 98.7 | 96.6 | 92.8 | 97.9 | 95.1 | 95.4 | 98.9 | 97.0 | 98.2 | 99.7 | 98.9 | 99.4 | 99.9 | 99.6 | 96.1 |
| | PMF | 94.9 | 98.4 | 96.5 | 92.8 | 97.7 | 95.0 | 96.3 | 99.1 | 97.6 | 98.5 | 99.7 | 99.1 | 99.5 | 100.0 | 99.7 | 96.4 |
| | HHEA | 89.9 | 95.5 | 92.5 | 89.7 | 95.2 | 92.2 | 68.1 | 78.8 | 73.2 | 77.0 | 86.0 | 81.1 | 85.8 | 92.2 | 88.7 | 82.1 |
| | Ours | 98.9 | 99.2 | 99.1 | 97.6 | 98.8 | 98.2 | 98.3 | 99.5 | 98.8 | 99.3 | 99.9 | 99.6 | 99.8 | 100.0 | 99.9 | 98.8 |
| 20% DNC | EVA | 67.4 | 76.2 | 71.6 | 17.9 | 21.4 | 19.7 | 64.2 | 78.9 | 70.8 | 72.6 | 85.7 | 78.5 | 88.0 | 95.2 | 91.3 | 62.0 |
| | MCLEA | 89.0 | 95.2 | 91.8 | 88.8 | 95.8 | 92.0 | 91.5 | 97.0 | 94.0 | 95.6 | 98.8 | 97.0 | 97.8 | 99.6 | 98.6 | 92.5 |
| | XGEA | 56.1 | 67.3 | 61.7 | 60.1 | 71.4 | 65.5 | 89.5 | 96.4 | 92.6 | 92.4 | 98.2 | 95.0 | 96.6 | 98.9 | 97.6 | 78.9 |
| | MEAformer | 93.8 | 97.6 | 95.6 | 91.8 | 97.2 | 94.3 | 95.5 | 98.5 | 96.8 | 98.3 | 99.6 | 98.9 | 99.4 | 99.9 | 99.7 | 95.7 |
| | UMAEA | 90.3 | 96.5 | 93.1 | 86.8 | 95.1 | 90.5 | 94.1 | 98.2 | 95.9 | 97.2 | 99.4 | 98.2 | 98.8 | 99.9 | 99.3 | 93.5 |
| | PMF | 92.2 | 96.9 | 94.3 | 90.9 | 96.3 | 93.4 | 94.8 | 98.1 | 96.3 | 97.6 | 99.3 | 98.3 | 99.2 | 99.9 | 99.5 | 94.9 |
| | HHEA | 87.6 | 93.8 | 90.5 | 89.3 | 94.6 | 92.1 | 66.1 | 75.9 | 70.8 | 72.6 | 81.9 | 77.0 | 83.5 | 90.2 | 86.6 | 79.8 |
| | Ours | 98.3 | 98.9 | 98.6 | 97.5 | 98.7 | 98.1 | 97.6 | 99.1 | 98.3 | 99.1 | 99.9 | 99.5 | 99.8 | 100.0 | 99.9 | 98.5 |
| 50% DNC | EVA | 2.7 | 3.8 | 3.4 | 0.0 | 0.1 | 0.2 | 17.5 | 31.7 | 24.2 | 18.4 | 33.2 | 25.2 | 15.3 | 30.5 | 22.4 | 10.8 |
| | MCLEA | 78.9 | 88.3 | 83.2 | 75.9 | 88.1 | 81.5 | 84.5 | 91.7 | 87.8 | 88.7 | 94.7 | 91.4 | 89.5 | 97.5 | 95.4 | 84.3 |
| | XGEA | 50.3 | 60.3 | 55.3 | 34.8 | 44.5 | 39.8 | 71.3 | 86.4 | 78.0 | 70.1 | 85.5 | 77.0 | 88.7 | 95.9 | 91.9 | 63.0 |
| | MEAformer | 91.9 | 96.7 | 94.1 | 91.9 | 96.8 | 94.1 | 93.4 | 97.3 | 95.2 | 97.3 | 99.1 | 98.1 | 99.1 | 99.9 | 99.5 | 94.7 |
| | UMAEA | 87.0 | 94.4 | 90.4 | 85.7 | 93.9 | 89.4 | 91.4 | 96.7 | 93.8 | 95.9 | 98.8 | 97.2 | 98.1 | 99.6 | 99.1 | 91.6 |
| | PMF | 86.9 | 93.9 | 90.0 | 87.6 | 94.4 | 90.7 | 92.2 | 96.5 | 94.2 | 96.1 | 98.8 | 97.3 | 98.6 | 99.6 | 99.1 | 92.3 |
| | HHEA | 86.2 | 92.8 | 89.2 | 84.2 | 92.1 | 87.8 | 56.8 | 71.3 | 63.7 | 70.5 | 82.2 | 75.9 | 76.9 | 86.1 | 81.1 | 74.9 |
| | Ours | 97.7 | 98.3 | 98.0 | 97.0 | 98.2 | 97.6 | 96.3 | 98.1 | 97.2 | 98.7 | 99.7 | 99.1 | 99.7 | 100.0 | 99.8 | 97.9 |

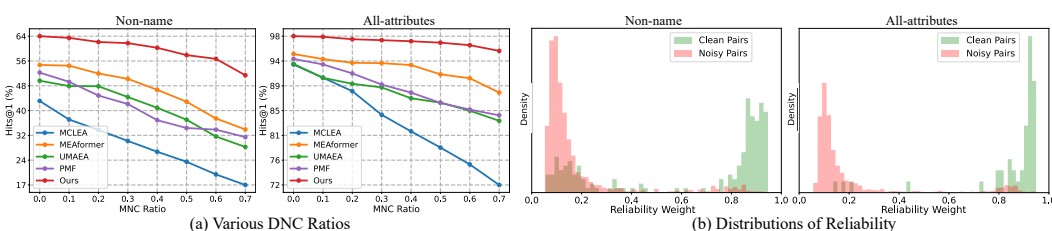

(a) Various DNC Ratios    (b) Distributions of Reliability

Figure 3: Analytic studies of various DNC ratios and reliability in Eq. 1.

in the paper. To further evaluate the robustness toward DNC, we manually inject noise to conduct more comprehensive evaluations by following the widely-adopted strategies in the noisy correspondence/label learning community (Natarajan et al., 2013; Huang et al., 2021). Specifically, the artificial noise are injected in the following three aspects: i) *entity-entity NC*: one entity in an aligned entity pair is randomly replaced with a different entity; ii) *entity-attribute NC*: a visual or textual attribute is randomly reassigned to a different entity; iii) *attribute-attribute NC*: visual attributes are perturbed with Gaussian noise, while textual attributes are corrupted via random character replacements. The artificial noise levels are set as $20\%$ and $50\%$ in our experiments, which represents the proportion of corrupted E-E/E-A/A-A pairs.

## 3.2 COMPARISONS WITH STATE-OF-THE-ARTS

In this section, we compare our method RULE with seven state-of-the-art MMEA methods under the Dual-level Noisy Correspondence setting, including EVA (Liu et al., 2021), MCLEA (Lin et al., 2022b), XGEA (Xu et al., 2023), MEAformer (Chen et al., 2023a), UMAEA (Chen et al., 2023b), PMF (Huang et al., 2024a), and HHEA (Jiang et al., 2024). Following (Chen et al., 2023a; Huang et al., 2024a; Xu et al., 2023), we conduct experiments under two widely-adopted evaluation protocols: *Non-name setting* denotes all attributes except for the entity name are used, while *All-attributes setting* includes all available modalities. For fair comparisons, we adopt the same backbone (*i.e.*, CLIP) for all baselines and our method. For more results on different backbones, please refer to Appendix G.11.

As shown in Tables 1-2, we could have the following conclusions: i) existing methods face substantial performance degradation as noise increases, highlighting their vulnerability to noisy correspondences. In contrast, RULE outperforms all baselines across different datasets and noise settings, demonstrating superior robustness against DNC; ii) even without any manually-injected noise, RULE still achieves performance gains compared to existing methods, as the real-world MMEA datasets contain a

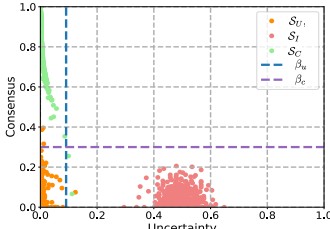

Figure 4: Quantitative analysis of the uncertainty and consensus on the name attribute.

Table 3: Ablation study of the various modules in the train and test stages.

| Stage | Setting | Non-name | | | All-attributes | | |
|---|---|---|---|---|---|---|---|
| | | H@1 | H@5 | MRR | H@1 | H@5 | MRR |
| Train | w/o DRL | 31.6 | 45.9 | 38.6 | 82.3 | 90.4 | 86.0 |
| | w/o DRF | 50.4 | 66.2 | 57.6 | 93.4 | 97.4 | 95.2 |
| | Only Unc. | 53.5 | 67.8 | 60.2 | 93.6 | 97.4 | 95.4 |
| | Only Cons. | 48.3 | 60.3 | 54.3 | 87.7 | 93.2 | 90.4 |
| Test | w/o DRF | 52.4 | 66.2 | 59.0 | 95.1 | 97.9 | 96.3 |
| | w/o TTR | 56.5 | 68.6 | 62.3 | 94.0 | 97.7 | 95.7 |
| | MLLM Enhance | 56.6 | 69.0 | 62.4 | 97.6 | 98.2 | 97.9 |
| Both | Default | 58.2 | 69.7 | 63.6 | 97.7 | 98.3 | 98.0 |

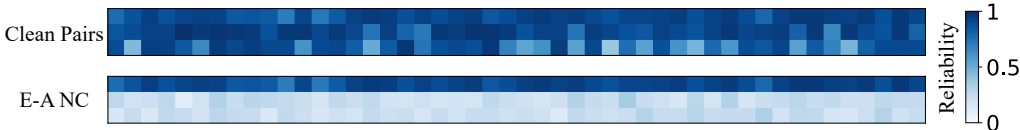

Figure 5: Visualization of the reliability for clean entity pairs and those with manually injected E-A NC in the image and name attributes. From top to bottom, the three rows denote the structure, image, and name attributes, respectively.

considerable number of DNC. To further verify the effectiveness of RULE, we conduct experiments under the manually-injected noise ratio from 0.0 to 0.7. As shown in Fig. 3 (a), RULE not only achieves higher performance across all noise levels but also exhibits significantly slower performance degradation, which further confirms the robustness of RULE against DNC.

## 3.3 ANALYSIS AND ABLATION STUDY

In this section, we conduct analysis and ablation studies on the ICEWS-WIKI dataset.

**Analysis Studies on Uncertainty and Consensus.** As discussed in Section 2.2, the estimated reliability plays a key role in identifying DNC. To better understand its behavior, we visualize the reliability distribution of all training entity pairs. As shown in Fig. 3(b), clean pairs are concentrated on the right side of the plot (indicating high reliability), while noisy pairs are predominantly on the left (indicating low reliability). This confirms that the proposed reliability serves an effective indicator for distinguishing clean and noisy pairs. To further explore how the proposed uncertainty and consensus behave under noise, we construct subsets $\mathcal{S}_U$ and $\mathcal{S}_I$ by injecting synthetic noise and randomly shuffling the name attributes of the raw set $\mathcal{S}_C$. As illustrated in Fig. 4, uncertainty and consensus principles successfully separate the three subsets, which supports the design of our tailored loss strategies in Eq. 11.

**Effectiveness of Robust Fusion**. To qualitatively study the effectiveness of RULE in handling entity-attribute noise, we visualize the reliability in Eq. 1 during the fusion process. As shown in Fig.5, correctly associated attributes are assigned high reliability scores, while noisy or irrelevant attributes receive significantly lower scores. This behavior confirms that RULE effectively suppresses the influence of unreliable attributes during fusion, thereby enhancing robustness against entity-attribute noise.

**Ablation studies.** To verify the effectiveness of each component in our framework, we conduct ablation experiments on the modules involved in both training and test-time phases. According to the results in Table 3, one could have the following conclusions. First, during training phase, both the "Only Unc." variant (which applies the uncertainty-guided loss in Eq. 18) and the "Only Cons." variant (which uses a consensus-based MSE loss) outperform the baseline "w/o DRL" (which uses only a standard MSE loss). This demonstrates the effectiveness of our proposed Dually Robust Learning mechanism in handling noisy correspondence. Second, during the test phase, the TTR module significantly improves alignment performance by uncovering latent semantic connections. In particular, the comparison between the "MLLM Enhance" (which only uses rethinking scores in Eq. 16) and "w/o TTR" settings shows that combining rethinking scores with prior similarity scores leads to complementary effects, resulting in improved robustness and accuracy. Third, the Dually

Robust Fusion (DRF) module effectively mitigates the influence of intra-entity NC. Its inclusion enhances performance in both the training and testing stages.

## 4 CONCLUSION

In this paper, we study a new problem in MMEA, *i.e.*, Dual-level Noisy Correspondence, which refers to the wrongly annotated intra-entity and inter-graph correspondences. To solve this problem, the proposed methods estimate the reliability of both the intra-entity and inter-graph correspondences and alleviate the negative impact of NC during the inter-graph discrepancy elimination and intra-entity attribution fusion. Beyond the training-time design, we employ a novel correspondence reasoning module to guarantee more accurate equivalent entity identification during inference. We believe this work might remarkably enrich the learning paradigm with noisy correspondence by simultaneously considering the noise across both training-time and test-time.

AUTHOR CONTRIBUTIONS

All authors contributed significantly to this work. Xi Peng and Mouxing Yang conceived the study, designed the RULE algorithm, refined the manuscript, and supervised the project. Haobin Li co-designed and implemented the RULE algorithm, conducted the baseline evaluations, and drafted the manuscript. Yijie Lin and Peng Hu analyzed the experimental results and contributed to the formulation of the manuscript. All authors reviewed and approved the final version.

ACKNOWLEDGMENTS

This work was supported in part by NSFC under Grant 624B2099; in part by the Fundamental Research Funds for the Central Universities under Grant CJ202303; in part by Sichuan Science and Technology Planning Project under Grant 24NSFTD0130; and in part by Fundamental and Interdisciplinary Disciplines Breakthrough Plan of the Ministry of Education of China under Grant JYB2025XDXM610.

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

# APPENDIX

## A  THE USE OF LARGE LANGUAGE MODELS (LLMS)

Yes, we employ LLMs to aid and polish writing. Specifically, LLMs are used for language refinement in the Introduction, Method, Experiments sections, and so on. Moreover, the multi-modal large language model (MLLM) acts as one of the key parts for the test-time correspondence reasoning module (see Section 2.5) in our method.

## B  NOISE STATISTICS IN REAL-WORLD BENCHMARKS

In this section, we elaborate on the necessity of addressing the DNC challenge in real-world scenarios and discuss the underlying reasons behind the formation of the DNC.

### B.1  STATISTICS ANALYSIS

As discussed in the manuscript, real-world benchmarks implicitly contain both intra-entity and inter-graph noisy correspondences. To support this claim, we conducted an additional observational study to analyze the types and distributions of noise present in the datasets. Specifically, we randomly sample 1,000 entity pairs from the training set in the ICEWS-WIKI and ICEWS-YAGO benchmarks and conduct manual statistical analysis. After that, we categorize each entity pair into one of four types, *i.e.*, "Clean", "E-E NC", "E-A NC" or "A-A NC". As shown in Fig. 6, one could have the following observation and conclusions:

- Even after careful manual annotation, a considerable amount of DNC still exists in real-world datasets, *i.e.*, more than 50% of entity pairs in ICEWS benchmarks suffer from the DNC challenge.
- Among the three types of NC, entity-attribute NC and attribute-attribute NC account for a large proportion (*e.g.*, over 40% in ICEWS benchmarks). Compared with the entity-entity NC, these two types of NC are more fine-grained, which makes it difficult to uncover manually. It is worth emphasizing that although several NC-oriented methods have been proposed, nearly all of them overlook such non-negligible fine-grained noise in MMEA, further highlighting the necessity of addressing the DNC challenge.

As mentioned above, although the two commonly-used MMEA datasets are small in scale (*i.e.*, containing up to 30K entity pairs) and carefully annotated by humans, they still exhibit non-negligible DNC cases. Such a DNC challenge would become even more severe in large-scale knowledge graphs such as Wikidata, DBpedia, and YAGO datasets (*i.e.*, containing over 10M-100M entities). We believe that the DNC challenge is one of the key reasons why large-scale datasets are currently lacking in the MMEA field. The proposed RULE framework offers a promising solution toward achieving entity alignment on larger-scale datasets with the DNC challenge.

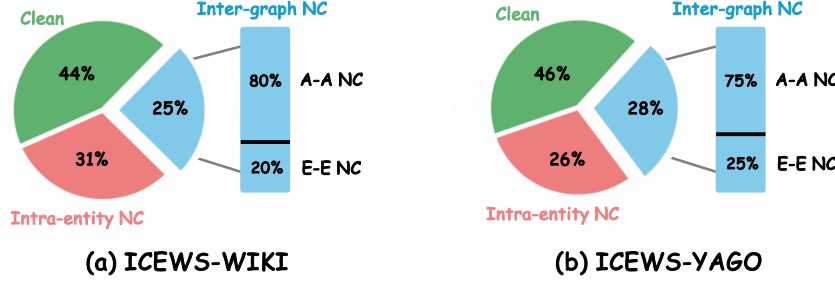

(a) ICEWS-WIKI  (b) ICEWS-YAGO

Figure 6: Distribution analysis of DNC noise in ICEWS-WIKI and ICEWS-YAGO benchmarks.

### B.2  THE UNDERLYING REASON BEHIND DNC

To investigate the underlying causes of DNC, we analyze the formation of intra-entity NC and inter-graph NC from two perspectives, *i.e.*, knowledge graph construction and cross-graph annotation.

**Knowledge Graph Construction:** The construction of existing knowledge graphs (Piscopo & Simperl, 2018) (*e.g.*, Wikidata, Freebase, and YAGO) heavily relies on crowdsourcing (Paulheim, 2016), *i.e.*, user editing on collaborative platforms. However, as most users lack expert knowledge, their edits are prone to errors such as incorrect data entry and conceptual misunderstandings, thereby inevitably introducing incorrect attributes within an entity. As a result, the editing errors in crowdsourcing would lead to intra-entity NC.

**Cross-graph Annotation:** In real-world scenarios, most MMEA datasets are annotated using semi-automated methods, which would inevitably introduce the inter-graph NC. Specifically, to construct entity pairs between the ICEWS and WIKI knowledge graphs, Jiang et al. (2024) employs the name attributes from the ICEWS graph as queries and uses the Wikidata API to retrieve the most relevant matches. As a result, entity-entity pairs are established between the queries and the retrieved results. However, such a process inevitably introduces inter-graph E-E NC due to the following reasons: i) some entities exist in ICEWS but do not appear in Wikidata, making it impossible to establish correct correspondences; ii) even some entities have highly similar name attributes, they could be very obvious mismatches. For instance, as shown in Fig. 1(a) in the manuscript, the movie entity "Mr. & Mrs. Smith" may be mistakenly aligned with the real-life actor couple "Will Smith and Mrs. Smith." In this case, despite the similarity in names, the two refer to totally different entities with distinct multi-modal attributes. In other words, entities considered "neighbors" during cross-graph annotation (*e.g.*, retrieved via name-based lookups), may not actually be relevant in real-world scenarios. Such fine-grained errors require expert knowledge to distinguish, and may be undetected even during manual filtering. Both of the above reasons would lead to obvious mistakes, where the aligned entity pairs are totally unrelated. It is worth noting that such obvious mistakes prove the reasonableness of constructing incorrect correspondences to simulate real-world noise in our experiments.

Moreover, once the inter-graph entity pairs are associated, their corresponding attributes could be treated as matched. Therefore, as the by-product of the entity-attribute and entity-entity correspondences establishment, the association between inter-graph attributes might be misled due to the aforementioned two kinds of annotation errors. In other words, E-A NC and E-E NC would further propagate to inter-graph attribute pairs, leading to noisy A-A correspondences. Beyond explicit annotation errors in A-A pairs, some attributes may contain textual typos or visual ambiguity (*e.g.*, noise under low-light conditions) in real-world scenarios, which would also lead to inter-graph A-A NC.

## C    RELATED WORK

In this section, we briefly review two topics related to this work, *i.e.*, multi-modal entity alignment and learning with noisy correspondence.

### C.1    MULTI-MODAL ENTITY ALIGNMENT

Multi-modal entity alignment aims to eliminate the discrepancy between heterogeneous MMKGs, so that the equivalent entities from various MMKGs could be identified. Towards achieving this goal, numerous MMEA approaches have been proposed, which typically involve the following two-stage pipeline, namely, fuse the intra-entity attributes to form the representation for entities and perform cross-graph alignment on the paired attributes and entities. According to their primary focus, most existing approaches could be broadly grouped into two categories: i) fusion-centric methods (Chen et al., 2023a; Huang et al., 2024a), which assign different weights according to the importance of each attribute-specific attribute during the fusion stage. ii) alignment-centric methods (Li et al., 2023; Lin et al., 2022b; Chen et al., 2024), which mitigate inter-graph discrepancy by maximizing the similarity between associated pairs while minimizing that of mismatched ones across MMKGs during the alignment stage.

Among the existing approaches, REA (Pei et al., 2020) is the most relevant to our work, while having the following remarkable differences. First, REA focuses on inter-graph uni-modality entity alignment, whereas our work tackles multi-modal entity alignment, which involves mitigating discrepancies not only between heterogeneous graphs but also across multi-modal attributes. Second, REA only considers misalignment in entity-entity pairs, while our work comprehensively reveals and studies the noisy correspondence at dual levels, namely, both the intra-entity and inter-graph. Third, unlike REA solely concentrates on training-time robustness, our method further improves test-time robustness, facilitating more precise cross-graph equivalent entity identification.

### C.2    LEARNING WITH NOISY CORRESPONDENCE

Noisy correspondence refers to inherently irrelevant or relevant samples that are wrongly regarded as associated (*a.k.a*, false positive) or unassociated (*a.k.a*, false negative), which is first revealed and studied by (Huang et al., 2021; Yang et al., 2021). Considering that numerous applications require paired data as input, including but not limited to visual instruction tuning (Huang et al., 2025), vision-language pre-training (Huang et al., 2024b), object re-identification (Yang et al., 2022), and graph matching (Lin et al., 2023), how to learn with noisy correspondence rooted in data pairs has emerged as a new research direction, drawing increasing attention from both academia and industry.

In this paper, we focus on mitigating the negative impact of the noisy positive correspondence issue. Unlike most existing noisy correspondence studies that tackle the errors in correspondence of a specific level (e.g., image-to-sample or pixel-to-pixel), this work delves into the multi-modal entity alignment task and reveals the specific dual-level noisy correspondence (DNC) problem for the first time. In brief, DNC refers to the noisy correspondence involving in the entity-attribute, entity-entity, and attribute-attribute pairs, which misleads the multi-modal fusion and cross-graph alignment processes. Therefore, it is desirable to customize a specific approach for MMEA with the DNC problem.

## D  EVIDENTIAL LEARNING

In this section, we provide more details about the dually robust objective function in Eq. 9.

Following (Sensoy et al., 2018; Han et al., 2022), the Dirichlet distribution is parameterized by $\boldsymbol{\alpha}_i = [\alpha_{i1}, \alpha_{i2}, \cdots, \alpha_{iK}]$ and the probability density function in Eq. 11 is given by,

$$D(\boldsymbol{p}_i \mid \boldsymbol{\alpha}_i) = \begin{cases} \frac{1}{B(\boldsymbol{\alpha}_i)} \prod_{j=1}^{\tilde{N}} p_{ij}^{\alpha_{ij}-1} & \text{for } \boldsymbol{p}_i \in \mathcal{S}^{\tilde{N}}, \\ 0 & \text{otherwise}, \end{cases} \tag{17}$$

where $B(\boldsymbol{\alpha}_i)$ is the multinomial beta function, and $\mathcal{S}^{\tilde{N}}$ is the $K$-dimensional unit simplex.

With the above derivation, MMEA aims to guide the query probability $\boldsymbol{p}_i$ to approach the annotated correspondence $\boldsymbol{y}_i$. To achieve this, the uncertainty-based loss could be formulated as follows,

$$\begin{aligned} \mathcal{L}_U(\boldsymbol{\alpha}_i, \boldsymbol{y}_i) &= \int \|\boldsymbol{y}_i - \boldsymbol{p}_i\|_2^2 \frac{1}{B(\boldsymbol{\alpha}_i)} \prod_{j=1}^{\tilde{N}} p_{ij}^{\alpha_{ij}-1} d\mathbf{p}_i \\ &= \sum_{j=1}^{\tilde{N}} \left[ (y_{ij} - \mathbb{E}[p_{ij}])^2 + \text{Var}(p_{ij}) \right] \\ &= \sum_{j=1}^{\tilde{N}} \left( y_{ij} - \frac{\alpha_{ij}}{Q_i} \right)^2 + \frac{\alpha_{ij}(Q_i - \alpha_{ij})}{Q_i^2(Q_i+1)}. \end{aligned} \tag{18}$$

where $\mathbb{E}[p_{ij}]$ and $\text{Var}(p_{ij})$ are the expected value and the variance of $p_{ij}$, respectively. Following (Sensoy et al., 2018), the expected probability $\mathbb{E}[p_{ij}]$ could be estimated by $\frac{\alpha_{ij}}{Q_i}$. Intuitively, Eq. 18 encourages higher evidence for correct correspondence compared to mismatched ones, but also prevents excessive optimization when the overall evidence is limited. In other words, the probability $\frac{\alpha_{ij}}{Q_i}$ is proportional to the total evidence $Q_i$, thus limiting overconfidence in noisy correspondences. Specifically,

**Theorem 2.** *The uncertainty-aware probability $\frac{\alpha_{ij}}{Q_i}$ is upper bounded by $\frac{Q_i-K+1}{Q_i}$, which is proportional to $Q_i$, i.e.,*

$$\frac{\alpha_{ij}}{Q_i} \leq \frac{Q_i - K + 1}{Q_i} \propto Q_i. \tag{19}$$

*Proof.* According to the definition of evidence in Eq. 2, each correspondence satisfies $\alpha_{il} \geq 1$ for all $l \neq j$. Therefore, $Q_i$ is lower-bounded by

$$Q_i = \alpha_{ij} + \sum_{l \neq j} \alpha_{il} \geq \alpha_{ij} + (K-1). \tag{20}$$

Rearranging the inequality yields,

$$\alpha_{ij} \leq Q_i - (K-1). \tag{21}$$

Then, dividing both sides by $Q_i$, we obtain the desired upper bound,

$$\frac{\alpha_{ij}}{Q_i} \leq \frac{Q_i - K + 1}{Q_i}, \tag{22}$$

where the upper bound is proportional to $Q_i$, *i.e.*, $\frac{Q_i-K+1}{Q_i} \propto Q_i$. $\qquad\square$

## E  PROOF OF THEOREM 1

In this section, we present detailed proofs for Theorem 1 in the main paper.

**Theorem 3.** *A low uncertainty $u_i$ does not necessarily imply that the highest belief is assigned to the annotated correspondence, i.e., $y_{ij} = 1$,*

$$z_i \text{ with low } u_i \not\Rightarrow \arg\max \boldsymbol{b}_i = \arg\max \boldsymbol{y}_i. \tag{23}$$

*Proof.* We will prove the theorem by contradiction. Suppose that for *all* evidence vectors with low uncertainty $u_i$, the highest belief is assigned to the annotated correspondence, i.e.,

$$\forall \boldsymbol{e}_i \text{ with low } u_i, \ \arg\max \boldsymbol{b}_i = \arg\max \boldsymbol{y}_i. \tag{24}$$

Let us now consider the two evidence vectors $\boldsymbol{e}_i^{(1)}$ and $\boldsymbol{e}_i^{(2)}$ defined as:

$$e_{ij}^{(1)} = \begin{cases} Q_i, & \text{if } j = j_1, \\ 0, & \text{otherwise,} \end{cases} \qquad e_{ij}^{(2)} = \begin{cases} Q_i, & \text{if } j = j_2, \\ 0, & \text{otherwise,} \end{cases} \tag{25}$$

where $j_1 \neq j_2$, and $Q_i > 0$ is fixed. According to Eq. 3, both evidence vectors yield the same total evidence $Q_i$ and hence the same uncertainty $u_i$. However, $\boldsymbol{e}_i^{(1)}$ suggests $(i, j_1)$ is the most probable correspondence, while $\boldsymbol{e}_i^{(2)}$ suggests $(i, j_2)$ as the most plausible option. However, only one of them might indicate the correct correspondence. Such an example contradicts the assumption that low uncertainty invariably leads to the highest belief being assigned to the annotated correspondence $j$. Consequently, the initial assumption is invalid. □

## F  MORE IMPLEMENTATION DETAILS

### F.1  DATASETS

We place the detailed statistics of the five MMEA datasets used in our experiments in Table 4. Note that, not every entity is paired with image-type attributes.

Table 4: Details of MMEA datasets. "E-E pairs" indicates the number of cross-graph entity-entity correspondences.

| Dataset | KG | Entity | Triples | Numeric | Image | E-E pairs |
|---|---|---|---|---|---|---|
| DBP15K$_{\text{ZH-EN}}$ | ZH (Chinese) | 19388 | 70414 | 8111 | 15912 | 15000 |
| | EN (English) | 19572 | 95142 | 7173 | 14125 | 15000 |
| DBP15K$_{\text{JA-EN}}$ | JA (Japanese) | 19814 | 77214 | 5882 | 12739 | 15000 |
| | EN (English) | 19600 | 93484 | 6066 | 13741 | 15000 |
| DBP15K$_{\text{FR-EN}}$ | FR (French) | 19661 | 105998 | 4547 | 10599 | 15000 |
| | EN (English) | 19993 | 115722 | 6422 | 13858 | 15000 |
| ICEWS-WIKI | ICEWS | 11047 | 3527881 | - | 33341 | 5058 |
| | WIKI | 15896 | 198257 | - | 47688 | 5058 |
| ICEWS-YAGO | ICEWS | 26863 | 4192555 | - | 80589 | 18824 |
| | YAGO | 22734 | 107118 | - | 68202 | 18824 |

### F.2  NETWORKS

In this section, we describe the network architectures used for encoding multi-modal attributes. For the name and image attributes, we first employ the pre-trained CLIP ViT-L/14 (Radford et al., 2021) model to obtain the features. Note that, all the experiments are conducted on the same visual and textual features for fair comparison. Following (Chen et al., 2023a), we employ trainable fully connected layers to project the image and name attributes into embedding vectors. Missing image attributes are handled by assigning zero vectors as their embeddings.

For the DBP15K benchmarks, the multi-modal attributes include structure, relation, numerical value, image, name, and character attributes. Following (Xu et al., 2023), we adopt cross-modal graph

networks to encode the structure and relation attributes. For the numerical attribute, we employ the Bag-of-Words (BoW) model to encode the relations as fixed-length vectors. Besides, the character attributes are derived from character bigrams of the translated entity names.

For the ICEWS benchmarks, the multi-modal attributes include structure, image, and name attributes. Following (Chen et al., 2023a), we utilize Graph Attention Networks GAT to reduce the computational cost of encoding structural information, as the number of relation triplets in the ICEWS benchmark is significantly larger than that of the DBP15K benchmark.

### F.3 DETAILS OF DUALLY ROBUST FUSION MODULE

To achieve robustness against intra-entity E-A NC, we propose a dually robust fusion that assigns low weights to unreliable attributes. Here, we provide more details on how to construct the initial subset for the greedy strategy in Eq. 7. Although some attributes are often weakly associated or even irrelevant to the entity, we assume that most attributes should be correctly associated, otherwise, the cross-graph correspondence would not hold. Based on this assumption, we adopt the following strategy for constructing the initial subset $\pi_0$,

$$|\pi_0| \geq \begin{cases} \left\lfloor \dfrac{M}{2} \right\rfloor + 1 & \text{if } M \geq 3 \\ 1 & \text{if } M \leq 2. \end{cases} \tag{26}$$

Such behavior ensures that the initial subset contains at least half or more of the multi-modal attributes.

Notably, the formulation of the attribute uncertainty $u_i^m$ and consensus $c_i^m$ relies on correct cross-graph entity-entity correspondence. However, the cross-graph correspondences are mismatched when E-E NC occurs, leading to unreliable $u_i^m$ and $c_i^m$. Thus, we employ DRF only to entities $z_i$ that satisfy

$$(1 - u_i) + c_i \geq 1. \tag{27}$$

In other words, DRF is adopted to entities with reliable correspondences, which could prevent erroneous fusion.

### F.4 EVALUATION METRIC

We evaluate multi-modal entity alignment using Hit@1 (Li et al., 2025; Lin et al., 2026) and Mean Reciprocal Rank (MRR) (Huang et al., 2024a):

- Hit@$k$ measures the percentage of cases where the ground-truth candidate appears in the top-$k$ retrieved entity.
- Mean Reciprocal Rank (MRR) assesses the average inverse rank of the correct entity.

### F.5 DETAILS OF TEST-TIME REASONING MODULE

In the manuscript, we propose a test-time reasoning module to capture the underlying connections between attribute pairs. Here, we provide more details on the MLLM rethinking presented in Eq. 16.

To reduce the computational cost of MLLM reasoning, we avoid rethinking attribute pairs that satisfy the conditions as follows,

$$\max(\boldsymbol{s}_i^m) \geq 0.2 \quad \lor \quad \max(\boldsymbol{s}_i^m) - \boldsymbol{s}_{ij}^m \geq 0.2, \ \forall \boldsymbol{s}_{ij}^m \neq \max(\boldsymbol{s}_i^m) \tag{28}$$

Such behavior not only reduces the overhead of certain attribute pairs, but also encourages reasoning on unreliable pairs and thus improves MMEA performance. Notably, for the missing image attributes in the DBP15K benchmark, the corresponding rethinking scores are set to zero.

Taking the entity pair $7125 - 26134$ as an example, we employ the following Chain-of-Thought for the Non-name and All-attributes setting.

For a given query entity $z_i$, the output results of the MLLM denotes as $\boldsymbol{o}_i = [o_{i1}; o_{i2}; \cdots, o_{i10}]$, where each similarity satisfies $0 \leq o_{ij} \leq 10$. After that, $\boldsymbol{o}_i$ is normalized as follow,

$$\hat{\boldsymbol{o}}_i = (\boldsymbol{o}_i - 5)/5. \tag{29}$$

Such normalization ensures that the range of the MLLM output is consistent with that of the similarity vector $s_i$ in Eq. 2. The $\hat{o}_i$ could then employed to derive the rethinking scores via Eq. 16, *i.e.*, $\oplus_{j \in \mathcal{T}_i^m} \left( \text{CoT} \left[ x_i^m, (x_j')^m, s_i^m \right] \right) = \hat{o}_i$.

---

**Chain-of-Thought for Non-name Setting**

- **Base Prompt:**
  - Help me align or match entities of different knowledge graphs according to the given images and prior retrieval results.

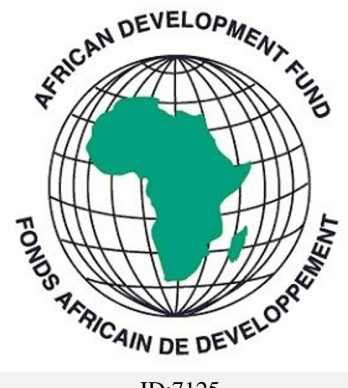 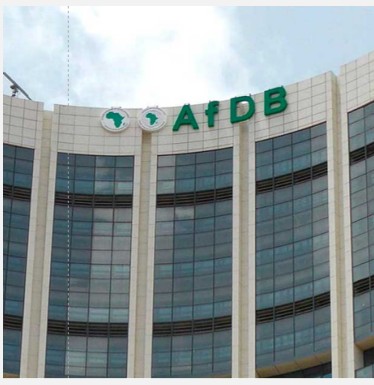

ID:7125                    ID:26134

- **Prior Results:**
  - Below are prior retrieval results focusing on visual similarity of the given images.
  - Candidate Entities List which may be aligned with QUERY Entity (ID:7125) are shown in the following list [Format: ID Similarity]:
    * 22646 0.40,
    * 22946 0.36,
    * 23688 0.30,
    * 26364 0.26,
    * 22619 0.20,
    * 26052 0.14,
    * 26134 0.10,
    * 26518 0.02.

- **Rethinking Image Similarity:**
  - The two provided images represent the query (ID:7125) and the candidate (ID:26134).
  - Please evaluate the probability that the QUERY and the CANDIDATE belong to the same entity **STEP BY STEP**:
  - 1. Rethink the visual similarities based on the prior retrieval results and the given images.
  - 2. Analyze the similarities of detailed visual contents between the provided images.
  - 3. Consider the underlying connections between the given images.
  - [Output Format]: [IMAGE SIMILARITY]= A out of 10, where A is in range [0,1,2,3,4,5,6,7,8,9,10], which represents the levels from VERY LOW to VERY HIGH. NOTICE: You MUST output strictly in this format: [IMAGE SIMILARITY]= A out of 10.

---

**Chain-of-Thought for All-attributes Setting**

- **Base Prompt:**
  - Help me align or match entities of different knowledge graphs according to the given names, images and prior retrieval results.

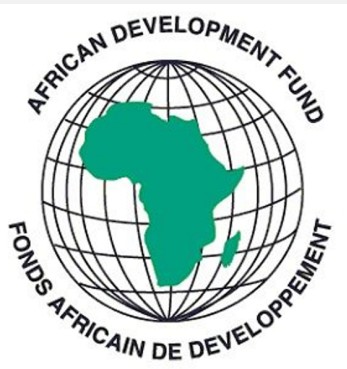

| ID:7125 Name:African Development Fund | ID:26134 Name:African Development Bank |

- **Prior Results:**
  - Below are prior retrieval results focusing on visual and textual similarity of the given images and names, respectively.
  - Candidate Entities List which may be aligned with QUERY Entity (ID:7125 Name:African Development Fund) are shown in the following list [Format: ID Name Similarity]:
    * 26364 African Development Bank 0.42,
    * 22619 Southern African Development Community 0.32,
    * 26052 Joseph Ki-Zerbo 0.24,
    * 22946 United Nations African Union Mission in Darfur 0.15,
    * 23688 OHADA 0.07,
    * 22646 Food and Agriculture Organization 0.05,
    * 26134 Jorg Asmussen -0.02,
    * 26518 Joyce Banda -0.08.

- **Rethinking Image Similarity:**
  - The two provided images represent the query (ID:7125 Name:African Development Fund) and the candidate (ID:26134 Name:African Development Bank).
  - Please evaluate the probability that the QUERY and the CANDIDATE belong to the same entity **STEP BY STEP**:
  - 1. Rethink the visual similarities based on the prior retrieval results and the given images.
  - 2. Analyze the similarities of detailed visual contents between the provided images.
  - 3. Consider the underlying connections between the given images.
  - [Output Format]: [IMAGE SIMILARITY]= A out of 10, where A is in range [0,1,2,3,4,5,6,7,8,9,10], which represents the levels from VERY LOW to VERY HIGH. NOTICE: You MUST output strictly in this format: [IMAGE SIMILARITY]= A out of 10.

- **Rethinking Name Similarity:**
  - The two provided names represent the query (ID:7125 Name:African Development Fund) and the candidate (ID:26134 Name:African Development Bank).
  - Based on the prior retrieval results and the given names, identify the similarities between the query entity and candidate entity.
  - [Output Format]: [NAME SIMILARITY]= B out of 10, where B is in range [0,1,2,3,4,5,6,7,8,9,10], which represents the levels from VERY LOW to VERY HIGH. NOTICE: You MUST output strictly in this format: [IMAGE SIMILARITY]= B out of 10.

## G    MORE EXPERIMENT RESULTS

In this section, we provide more experimental results of the proposed RULE. Unless otherwise specified, all experiments are conducted on the ICEWS-WIKI dataset under the 50% DNC setting.

### G.1    RESULTS UNDER THE E-E, E-A AND A-A NOISY CORRESPONDENCE

In the manuscript, we have carried out experiments under DNC settings. Here, we further provide more results under single-type NC scenarios, *i.e.*, E-E, E-A, and A-A NC. From the results in Table 5-6, RULE significantly outperforms all baselines across various settings and datasets.

Table 5: Comparisons with state-of-the-art methods on Non-name benchmarks under NC setting regarding the Hits and MRR metrics.

| Setting | Method | ICEWS-WIKI | | | ICEWS-YAGO | | | DBP15K$_{ZH-EN}$ | | | DBP15K$_{JA-EN}$ | | | DBP15K$_{FR-EN}$ | | | Avg. |
| | | H@1 | H@5 | MRR | H@1 | H@5 | MRR | H@1 | H@5 | MRR | H@1 | H@5 | MRR | H@1 | H@5 | MRR | H@1 |
|---|---|---|---|---|---|---|---|---|---|---|---|---|---|---|---|---|---|
| 50% E-E NC | EVA | 4.4 | 6.9 | 5.7 | 0.1 | 0.1 | 0.2 | 17.4 | 30.6 | 23.6 | 21.9 | 34.9 | 28.1 | 18.9 | 31.1 | 24.9 | 12.5 |
| | MCLEA | 28.6 | 43.9 | 36.2 | 20.3 | 34.7 | 27.4 | 58.2 | 75.6 | 66.1 | 58.2 | 74.6 | 65.8 | 59.1 | 75.3 | 66.6 | 44.9 |
| | XGEA | 40.0 | 47.4 | 44.0 | 21.8 | 26.5 | 24.7 | 70.1 | 85.2 | 75.9 | 68.9 | 82.1 | 74.2 | 68.8 | 81.9 | 72.0 | 53.9 |
| | MEAformer | 46.0 | 62.6 | 53.6 | 33.6 | 48.3 | 40.7 | 68.9 | 83.1 | 75.3 | 65.0 | 80.5 | 72.1 | 66.8 | 81.4 | 73.5 | 56.1 |
| | UMAEA | 44.0 | 60.4 | 51.9 | 28.1 | 43.5 | 35.5 | 66.8 | 82.8 | 74.0 | 62.2 | 79.4 | 70.1 | 63.7 | 80.4 | 71.2 | 53.0 |
| | PMF | 38.6 | 53.9 | 46.0 | 32.6 | 45.4 | 38.8 | 68.4 | 83.3 | 75.2 | 67.1 | 81.8 | 73.8 | 67.5 | 82.1 | 74.2 | 54.8 |
| | HHEA | 45.6 | 59.1 | 52.1 | 37.0 | 48.5 | 42.6 | 47.7 | 56.2 | 52.0 | 49.3 | 56.8 | 53.3 | 52.2 | 59.4 | 56.0 | 46.4 |
| | Ours | **61.0** | **72.5** | **66.6** | **48.7** | **59.5** | **54.0** | **73.6** | **86.1** | **79.3** | **71.6** | **84.6** | **77.6** | **71.5** | **84.7** | **77.6** | **65.3** |
| 50% E-A NC | EVA | 5.5 | 9.4 | 7.7 | 0.2 | 0.9 | 0.9 | 64.6 | 83.4 | 76.0 | 67.8 | 86.1 | 76.0 | 69.1 | 87.8 | 77.2 | 41.4 |
| | MCLEA | 34.4 | 51.9 | 42.9 | 24.0 | 39.1 | 31.5 | 75.3 | 89.6 | 81.7 | 75.4 | 90.4 | 82.0 | 76.2 | 90.9 | 82.7 | 57.1 |
| | XGEA | 40.0 | 47.9 | 44.3 | 22.4 | 27.9 | 25.7 | 76.1 | 91.2 | 82.6 | 75.8 | 91.2 | 82.5 | 76.0 | 92.1 | 83.3 | 58.1 |
| | MEAformer | 53.1 | 68.6 | 60.4 | 37.4 | 52.8 | 44.8 | 75.8 | 91.1 | 82.5 | 75.8 | 91.8 | 82.9 | 76.6 | 92.6 | 83.7 | 63.7 |
| | UMAEA | 50.7 | 67.5 | 58.5 | 34.3 | 49.0 | 41.6 | 72.3 | 89.8 | 80.1 | 72.2 | 90.2 | 80.2 | 73.8 | 91.6 | 81.7 | 60.7 |
| | PMF | 44.8 | 59.4 | 51.9 | 34.9 | 48.5 | 41.5 | 74.3 | 89.7 | 81.1 | 74.4 | 90.4 | 81.5 | 76.0 | 92.3 | 83.2 | 60.9 |
| | HHEA | 49.9 | 63.7 | 56.4 | 35.8 | 48.9 | 42.1 | 46.4 | 60.8 | 53.6 | 48.1 | 61.2 | 54.6 | 52.4 | 64.0 | 58.3 | 46.0 |
| | Ours | **62.6** | **75.5** | **68.7** | **47.7** | **59.1** | **53.5** | **80.3** | **92.4** | **85.7** | **79.1** | **92.5** | **84.9** | **80.1** | **93.2** | **85.9** | **70.0** |
| 50% A-A NC | EVA | 29.3 | 41.1 | 35.1 | 5.5 | 10.3 | 8.1 | 70.2 | 86.4 | 77.5 | 72.3 | 89.1 | 79.8 | 73.6 | 90.4 | 80.9 | 50.2 |
| | MCLEA | 43.2 | 62.8 | 52.4 | 29.7 | 47.1 | 38.3 | 74.8 | 89.5 | 81.4 | 75.4 | 90.2 | 82.0 | 76.1 | 90.7 | 82.6 | 59.8 |
| | XGEA | 40.8 | 48.8 | 45.1 | 24.2 | 29.7 | 27.6 | 78.2 | 92.1 | 84.4 | 78.8 | 92.9 | 84.9 | 81.0 | 93.9 | 86.6 | 60.6 |
| | MEAformer | 53.4 | 69.1 | 60.9 | 34.7 | 51.1 | 42.6 | 81.9 | 93.5 | 87.0 | 81.4 | 94.2 | 87.0 | 82.1 | 94.4 | 87.5 | 66.7 |
| | UMAEA | 49.6 | 68.0 | 58.0 | 29.5 | 45.8 | 37.6 | 77.8 | 92.1 | 84.3 | 78.3 | 92.3 | 84.5 | 79.3 | 93.5 | 85.6 | 62.9 |
| | PMF | 51.9 | 67.3 | 59.4 | 37.1 | 52.5 | 44.8 | 80.7 | 93.1 | 86.2 | 81.1 | 93.4 | 86.6 | 82.7 | 94.5 | 88.0 | 66.7 |
| | HHEA | 48.6 | 63.3 | 55.8 | 37.0 | 50.2 | 43.4 | 46.6 | 61.1 | 53.8 | 47.3 | 59.7 | 53.5 | 50.1 | 62.4 | 56.3 | 45.9 |
| | Ours | **63.6** | **76.7** | **69.7** | **48.6** | **60.3** | **54.4** | **85.6** | **94.7** | **89.7** | **85.1** | **95.4** | **89.5** | **85.0** | **95.5** | **89.6** | **73.6** |

Table 6: Comparisons with state-of-the-art methods on All-attributes benchmarks under NC settings regarding the Hits and MRR metrics.

| Setting | Method | ICEWS-WIKI | | | ICEWS-YAGO | | | DBP15K$_{ZH-EN}$ | | | DBP15K$_{JA-EN}$ | | | DBP15K$_{FR-EN}$ | | | Avg. |
| | | H@1 | H@5 | MRR | H@1 | H@5 | MRR | H@1 | H@5 | MRR | H@1 | H@5 | MRR | H@1 | H@5 | MRR | H@1 |
|---|---|---|---|---|---|---|---|---|---|---|---|---|---|---|---|---|---|
| 50% E-E NC | EVA | 53.6 | 61.8 | 57.7 | 4.4 | 5.7 | 5.1 | 25.6 | 38.3 | 31.6 | 32.6 | 44.3 | 38.2 | 51.2 | 61.1 | 56.0 | 33.5 |
| | MCLEA | 85.0 | 92.7 | 88.5 | 86.4 | 94.5 | 90.1 | 89.2 | 94.9 | 91.8 | 94.3 | 97.7 | 95.8 | 98.1 | 99.6 | 98.8 | 90.6 |
| | XGEA | 57.4 | 68.8 | 62.9 | 57.0 | 68.9 | 62.7 | 87.8 | 95.2 | 91.2 | 91.6 | 97.0 | 94.1 | 96.6 | 99.0 | 97.7 | 78.1 |
| | MEAformer | 92.7 | 97.1 | 94.7 | 91.7 | 97.0 | 94.1 | 94.0 | 97.5 | 95.6 | 97.6 | 99.2 | 98.3 | 99.2 | 99.8 | 99.5 | 95.0 |
| | UMAEA | 90.1 | 96.4 | 93.0 | 87.7 | 95.4 | 91.2 | 92.6 | 97.3 | 94.7 | 96.5 | 98.9 | 97.6 | 98.5 | 99.7 | 99.1 | 93.1 |
| | PMF | 90.7 | 96.3 | 93.2 | 88.4 | 95.3 | 91.5 | 92.7 | 96.6 | 94.5 | 96.4 | 98.6 | 97.4 | 98.6 | 99.7 | 99.1 | 93.3 |
| | HHEA | 88.7 | 93.9 | 91.2 | 89.7 | 94.7 | 92.1 | 67.9 | 78.4 | 72.8 | 76.7 | 85.7 | 80.8 | 87.8 | 93.2 | 90.3 | 82.2 |
| | Ours | **98.4** | **99.0** | **98.7** | **97.5** | **98.6** | **98.8** | **98.0** | **96.4** | **98.2** | **98.6** | **97.3** | **99.1** | **99.6** | **99.9** | **99.8** | **98.3** |
| 50% E-A NC | EVA | 64.8 | 73.0 | 68.8 | 9.1 | 10.9 | 10.1 | 69.3 | 86.0 | 76.7 | 75.6 | 89.9 | 82.0 | 88.9 | 96.1 | 92.1 | 61.6 |
| | MCLEA | 87.3 | 93.9 | 90.2 | 85.7 | 94.5 | 89.7 | 93.5 | 97.9 | 95.5 | 96.7 | 99.4 | 97.9 | 98.9 | 99.7 | 99.3 | 92.4 |
| | XGEA | 51.4 | 61.6 | 56.4 | 42.3 | 53.2 | 47.7 | 86.6 | 95.6 | 90.6 | 89.5 | 96.9 | 92.8 | 98.1 | 99.7 | 98.8 | 73.6 |
| | MEAformer | 94.6 | 98.0 | 96.1 | 92.0 | 97.2 | 94.3 | 95.1 | 98.6 | 96.6 | 97.9 | 99.7 | 98.7 | 99.3 | 99.8 | 99.6 | 95.8 |
| | UMAEA | 91.1 | 96.6 | 93.6 | 86.9 | 95.1 | 90.6 | 93.1 | 98.0 | 95.3 | 96.5 | 99.2 | 97.7 | 98.6 | 99.8 | 99.2 | 93.2 |
| | PMF | 91.1 | 95.7 | 93.2 | 89.7 | 95.8 | 92.4 | 95.0 | 98.5 | 96.5 | 98.0 | 99.7 | 98.7 | 99.4 | 99.8 | 99.7 | 94.6 |
| | HHEA | 88.8 | 95.1 | 91.7 | 89.2 | 94.8 | 91.8 | 72.0 | 84.9 | 77.9 | 79.7 | 89.4 | 84.2 | 90.0 | 95.7 | 92.5 | 83.9 |
| | Ours | **98.4** | **98.9** | **98.7** | **97.5** | **98.6** | **98.7** | **98.0** | **97.7** | **99.3** | **99.4** | **98.4** | **99.4** | **99.7** | **100.0** | **99.8** | **98.6** |
| 50% A-A NC | EVA | 84.6 | 91.5 | 87.9 | 61.7 | 69.8 | 65.6 | 87.0 | 95.4 | 90.7 | 91.8 | 97.8 | 94.5 | 96.4 | 99.3 | 97.6 | 84.3 |
| | MCLEA | 92.2 | 97.2 | 94.5 | 90.9 | 97.1 | 93.7 | 92.5 | 97.9 | 94.9 | 95.9 | 99.2 | 97.3 | 97.8 | 99.6 | 98.7 | 93.9 |
| | XGEA | 56.3 | 67.8 | 61.8 | 87.2 | 94.6 | 90.6 | 88.3 | 96.1 | 91.8 | 91.8 | 97.2 | 94.2 | 94.5 | 98.0 | 96.0 | 83.6 |
| | MEAformer | 95.1 | 98.3 | 96.6 | 93.0 | 97.9 | 95.2 | 96.5 | 99.0 | 97.6 | 98.6 | 99.8 | 99.2 | 99.4 | 99.8 | 99.8 | 96.6 |
| | UMAEA | 92.1 | 97.3 | 94.4 | 86.7 | 95.1 | 90.5 | 95.2 | 98.9 | 96.9 | 97.9 | 99.6 | 98.7 | 99.0 | 99.7 | 99.4 | 94.2 |
| | PMF | 94.6 | 98.6 | 96.4 | 91.7 | 97.1 | 94.3 | 96.2 | 99.0 | 97.5 | 98.4 | 99.7 | 99.0 | 99.4 | 99.8 | 99.7 | 96.2 |
| | HHEA | 88.7 | 95.2 | 91.8 | 89.5 | 95.5 | 92.2 | 53.4 | 68.9 | 60.7 | 64.0 | 76.6 | 70.0 | 74.1 | 83.0 | 78.3 | 73.9 |
| | Ours | **98.4** | **99.0** | **98.7** | **97.5** | **98.7** | **98.9** | **98.1** | **98.2** | **99.5** | **99.6** | **99.9** | **99.6** | **99.8** | **100.0** | **99.9** | **98.7** |

## G.2 PARAMETER ANALYSIS

As mentioned in the manuscript, the hyperparameters in RULE include the trade-off parameter $\lambda$ in Eq. 9, the temperature $\tau$ in Eq. 2, and the threshold $\beta$ in Eq. 8. Here, we conduct a detailed parameter analysis to study their individual impacts and one could have the following conclusions: i) RULE exhibits stable performance when $\lambda$ is within the range of $[2e-5, 5e-4]$, $\tau$ within $[0.05, 0.2]$, and $\beta$ within $[0.2, 0.4]$. ii) when $\lambda$ is excessively large, the model would overemphasize the regularization term in Eq. 13, resulting in performance drop.

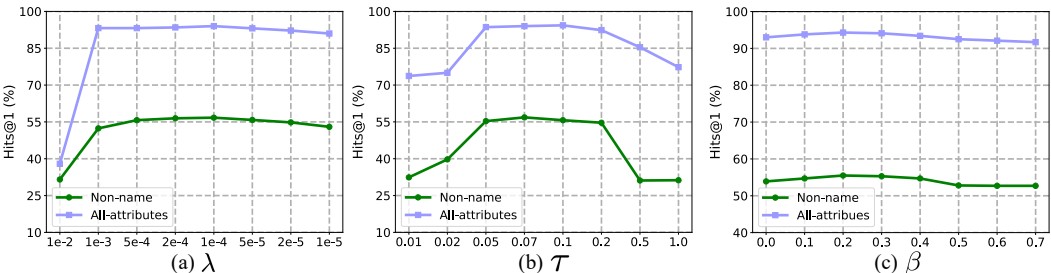

Figure 7: The parameter analysis of the trade-off parameter $\lambda$ in Eq. 9, the temperature $\tau$ in Eq. 2, and the threshold $\beta$ in Eq. 8.

## G.3 MORE ABLATION STUDIES

To further verify the effectiveness of the proposed RULE, we conduct additional ablation studies of the loss terms and the test-time recitification module. Specifically, we investigate the loss terms and the pair division module in Table 7, resulting in the following conclusions. First, the regularization loss penalizes the evidence of negative pairs, thereby enhancing the performance. Second, employing tailored strategies to either $\mathcal{S}_I$ or $\mathcal{S}_U$ would contribute to the performance improvement.

Table 7: Ablation study of the loss terms and the correspondence division module on the ICEWS-WIKI benchmark.

| Setting | | Non-name | | | All-attributes | | |
|---|---|---|---|---|---|---|---|
| $L_{DR}$ | $L_{Reg}$ | Hits@1 | Hits@5 | MRR | Hits@1 | Hits@5 | MRR |
| ✓ | | 55.8 | 68.3 | 61.8 | 93.0 | 97.7 | 96.1 |
| ✓ | ✓ | 56.5 | 68.6 | 62.3 | 94.0 | 97.7 | 95.7 |
| w/o $\mathcal{S}_I$ | | 55.6 | 67.8 | 61.5 | 93.5 | 97.4 | 95.4 |
| w/o $\mathcal{S}_U$ | | 54.8 | 67.9 | 61.1 | 93.5 | 97.2 | 95.2 |

As discussed in the manuscript, the test-time reasoning module would help to capture the underlying connection and thus boost the performance. Here, we carry out more ablation studies about the test-time reasoning module. From the results in Table 8-9, the proposed TTR module significantly improves performance on both the Non-name and All-attribute benchmarks. It is worth noting that, despite the missing image attributes in the DBP15K benchmark, RULE still achieves stable performance gains.

Table 8: Ablation study of TTR module on Non-name benchmarks under different DNC settings.

| Setting | Method | ICEWS-WIKI | | | ICEWS-YAGO | | | DBP15K$_{ZH-EN}$ | | | DBP15K$_{JA-EN}$ | | | DBP15K$_{FR-EN}$ | | |
|---|---|---|---|---|---|---|---|---|---|---|---|---|---|---|---|---|
| | | H@1 | H@5 | MRR | H@1 | H@5 | MRR | H@1 | H@5 | MRR | H@1 | H@5 | MRR | H@1 | H@5 | MRR |
| Inherent DNC | w/o TTR | 62.1 | 76.6 | 68.8 | 46.4 | 59.6 | 53.0 | 85.5 | 94.8 | 89.7 | 85.0 | 95.3 | 89.4 | 85.2 | 95.4 | 89.7 |
| | w TTR | 64.2 | 76.7 | 70.0 | 48.8 | 60.5 | 54.6 | 85.6 | 94.8 | 89.7 | 85.2 | 95.4 | 89.6 | 85.1 | 95.4 | 89.6 |
| 20% DNC | w/o TTR | 60.3 | 74.7 | 67.0 | 46.2 | 58.6 | 52.2 | 80.9 | 92.0 | 85.9 | 80.4 | 92.2 | 85.6 | 80.6 | 92.2 | 85.8 |
| | w TTR | 62.4 | 75.1 | 68.5 | 48.3 | 59.5 | 53.9 | 81.1 | 92.0 | 86.0 | 80.5 | 92.2 | 85.6 | 80.5 | 92.2 | 85.8 |
| 50% DNC | w/o TTR | 56.5 | 68.6 | 62.3 | 44.5 | 56.0 | 50.3 | 73.2 | 86.0 | 79.0 | 71.5 | 84.7 | 77.6 | 71.3 | 84.7 | 77.5 |
| | w TTR | 58.2 | 69.7 | 63.6 | 46.9 | 57.4 | 52.0 | 73.4 | 85.9 | 79.2 | 71.8 | 84.9 | 77.8 | 71.4 | 84.8 | 77.5 |

Table 9: Ablation study of TTR module on All-attributes benchmarks under different DNC settings.

| Setting | Method | ICEWS-WIKI | | | ICEWS-YAGO | | | DBP15K_ZH-EN | | | DBP15K_JA-EN | | | DBP15K_FR-EN | | |
|---|---|---|---|---|---|---|---|---|---|---|---|---|---|---|---|---|
| | | H@1 | H@5 | MRR | H@1 | H@5 | MRR | H@1 | H@5 | MRR | H@1 | H@5 | MRR | H@1 | H@5 | MRR |
| Inherent DNC | w/o TTR | 96.7 | 98.9 | 97.8 | 95.2 | 98.5 | 96.7 | 97.4 | 99.4 | 98.3 | 99.2 | 99.9 | 99.5 | 99.7 | 100.0 | 99.8 |
| | w TTR | 98.9 | 99.2 | 99.1 | 97.6 | 98.8 | 98.2 | 98.3 | 99.5 | 98.8 | 99.3 | 99.9 | 99.6 | 99.8 | 100.0 | 99.9 |
| 20% DNC | w/o TTR | 95.8 | 98.4 | 97.1 | 95.1 | 98.4 | 96.6 | 96.4 | 98.8 | 97.5 | 98.8 | 99.8 | 99.3 | 99.6 | 100.0 | 99.8 |
| | w TTR | 98.3 | 98.9 | 98.6 | 97.5 | 98.7 | 98.1 | 97.6 | 99.1 | 98.3 | 99.1 | 99.9 | 99.5 | 99.8 | 100.0 | 99.9 |
| 50% DNC | w/o TTR | 94.0 | 97.7 | 95.7 | 94.3 | 97.8 | 95.9 | 94.9 | 97.9 | 96.2 | 98.0 | 99.6 | 98.7 | 99.3 | 99.9 | 99.6 |
| | w TTR | 97.7 | 98.3 | 98.0 | 97.0 | 98.2 | 97.6 | 96.3 | 98.1 | 97.2 | 98.7 | 99.7 | 99.1 | 99.7 | 100.0 | 99.8 |

## G.4 MORE EXPERIMENTS UNDER VARIOUS DNC RATIOS

In the manuscript, we have carried out the experiments under various DNC ratios on the ICEWS-WIKI dataset. Here, we provide more experiments under various DNC ratios on the DBP15K_ZH-EN dataset. As shown in Fig 8, the proposed RULE significantly outperforms all the baselines.

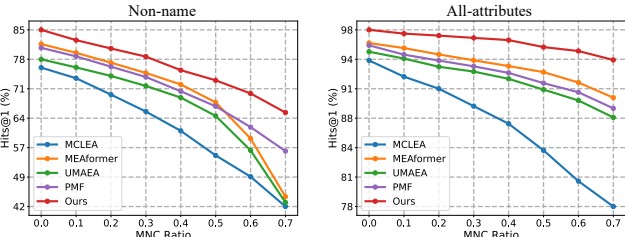

Figure 8: Performance with various DNC ratios on the DBP15K_ZH-EN dataset.

## G.5 MORE ANALYTIC STUDY OF DUALLY ROBUST FUSION

We conduct more analytic studies on the ICEWS-WIKI dataset to verify that the DRF achieves better performance by estimating the correct correspondence during the test time. From the results in Table 10, the estimated correspondence achieves higher accuracy compared to the vanilla fusion method, *i.e.*, simple concatenation. Consequently, RULE benefits from the estimated correspondence and the DRF module, resulting in better performance. Note that, different from previous fusion methods (Lin et al., 2024; 2022a), the proposed DRF further explores robust fusion on the reasoning scores output by the MLLMs.

Table 10: The analytic study of the estimated correspondence on the ICEWS-WIKI dataset.

| Setting | Non-name | | | All-attributes | | |
|---|---|---|---|---|---|---|
| | Hits@1 | Hits@5 | MRR | Hits@1 | Hits@5 | MRR |
| w/o DRF | 51.4 | 66.7 | 58.6 | 93.0 | 97.3 | 94.9 |
| Estimated | 55.3 | 65.0 | 60.2 | 94.1 | 97.4 | 95.7 |
| w DRF | 56.5 | 68.6 | 62.3 | 94.0 | 97.7 | 95.7 |

G.6 MORE ANALYTIC STUDY OF INTER-GRAPH LEARNING

As discussed in 3.3 and G.3, the proposed RULE improves the performance by adopting the tailored strategies to the divided subsets, *i.e.*, $\mathcal{S}_U$, $\mathcal{S}_I$ and $\mathcal{S}_C$. Here, we carry out more analytic studies about the statistics of the three kinds of pairs. From the results in Fig. 9 and Table 11, one could observe that the proposed correspondence division module effectively distinguishes the three subsets and thus boosts the performance. Different from previous discrepancy elimination methods (Gong et al., 2021; 2024; Li et al., 2024), the proposed DRL not only divides correspondences into positive and negative, but also further distinguishes different types of negatives and adapts the tailored strategy to mitigate the negative impacts of them.

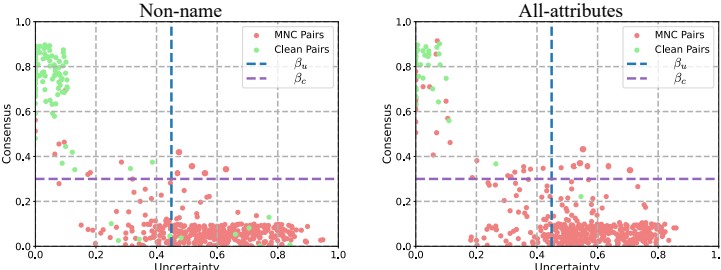

Figure 9: Quantitative analysis of the uncertainty and consensus on the integrated entity.

Table 11: Statistics of three kinds of pairs.

| Entity | $\mathcal{S}_C$ | $\mathcal{S}_I$ | $\mathcal{S}_U$ |
|---|---|---|---|
| Non-name | 14.8% | 33.9% | 51.3% |
| All-attributes | 14.1% | 37.0% | 48.9% |

G.7 RESULTS OF VARIOUS MLLM

To verify the generalizability of the TTR module, we conduct additional experiments on the MLLM with various architectures and parameter scales. Specifically, we evaluate the performance of the TTR module on Qwen2.5-VL models with 3B, 7B, and 72B parameters, as well as on LLaVA-1.6 (Liu et al., 2023) with 34B parameters. As shown in Table 12, RULE achieves consistent performance improvements across various model architectures and parameter scales, demonstrating the effectiveness of the proposed TTR module.

Table 12: Performance comparison of different MLLMs on Non-name and All-attributes settings.

| Methods | Non-name | | | All-attributes | | |
|---|---|---|---|---|---|---|
| | Hits@1 | Hits@5 | MRR | Hits@1 | Hits@5 | MRR |
| w/o TTR | 56.5 | 68.6 | 62.3 | 94.0 | 97.7 | 95.7 |
| Qwen2.5-VL 3B | 57.2 | 68.8 | 62.7 | 96.5 | 98.2 | 97.3 |
| Qwen2.5-VL 7B | 57.4 | 69.0 | 62.9 | 97.1 | 98.2 | 97.6 |
| Qwen2.5-VL 72B | 58.2 | 69.7 | 63.6 | 97.7 | 98.3 | 98.0 |
| LLaVA-1.6 34B | 57.0 | 68.8 | 62.6 | 95.5 | 97.9 | 96.6 |

G.8 COMPLEXITY ANALYSIS OF TEST-TIME CORRESPONDENCE REASONING

To assess the efficiency of the TTR module, we report the time cost and memory cost across various parameter scales of Qwen2.5-VL. Note that all experiments were conducted on the NVIDIA RTX 3090 GPUs. From the results in Table 13, one can observe that employing Qwen2.5-VL 3B or 7B for TTR leads to a considerable performance boost, with up to a 5× speedup. Furthermore, when resources are limited, RULE can be deployed without MLLMs. Even without TTR, RULE still significantly outperforms the strongest baselines, achieving 56.5% vs. 43.9% (best-performing

baseline) in the "Non-name" setting and 94.0% vs. 91.9% (best-performing baseline) in the "All-attributes" setting. Moreover, the lightweight solutions substantially reduce the memory cost as well.

Table 13: Experiment results for the complexity analysis on the ICEWS-WIKI dataset.

| Methods | Non-name | | All-attributes | | Memory Consumption |
|---|---|---|---|---|---|
| | H@1 | Time Cost | H@1 | Time Cost | |
| w/o TTR | 56.5 | 103 | 94.0 | 109 | 1GPU × ~16GB |
| Qwen2.5-VL 3B | 57.2 | 2122 | 96.5 | 1008 | 1GPU × ~8GB |
| Qwen2.5-VL 7B | 57.4 | 2690 | 97.1 | 1329 | 1GPU × ~8GB |
| Qwen2.5-VL 72B | 58.2 | 10043 | 97.7 | 4373 | 8GPU × ~20GB |

### G.9 RESULTS OF CROSS-ENTROPY IMPLEMENTATION

As discussed in (Sensoy et al., 2018), the uncertainty-based loss could be implemented using either the Mean Squared Error (MSE) loss or the Cross-Entropy (CE) loss. In Appendix D, we derive the uncertainty-based formulation based on MSE loss, which also serves as the foundation of the proposed dually robust loss in Eq. 11. Here, we derive the dually robust loss using the cross-entropy formulation as follows,

$$
\begin{aligned}
\mathcal{L}_{DR}(\boldsymbol{\alpha}_i, \hat{\boldsymbol{y}}_i) &= \mathbb{I}\left(u_i \leq \beta_u\right) \int \left[ \sum_{j=1}^{\tilde{N}} -\hat{y}_{ij} \log(p_{ij}) \right] D(\boldsymbol{p}_i \mid \boldsymbol{\alpha}_i)\, d\mathbf{p}_i \\
&= \mathbb{I}\left(u_i \leq \beta_u\right) \sum_{j=1}^{\tilde{N}} \hat{y}_{ij} \left(\psi(Q_i) - \psi(\alpha_{ij})\right),
\end{aligned}
\tag{30}
$$

where $\psi(\cdot)$ is the digamma function. To further investigate the effectiveness of the proposed DRL, we conduct additional experiments of the cross-entropy implementation based on Eq. 30. From the results in Table 14, both implementations based on MSE and CE losses achieve competitive performance compared to other baselines shown in Tables 1 and 2.

Table 14: Comparisons of RULE with MAE and CE objectives under the DNC setting.

| Objective | Setting | Non-name | | | All-attributes | | |
|---|---|---|---|---|---|---|---|
| | | Hits@1 | Hits@5 | MRR | Hits@1 | Hits@5 | MRR |
| MAE | Inherent DNC | 62.1 | 76.6 | 68.8 | 96.7 | 98.9 | 97.8 |
| | 20% DNC | 60.3 | 74.7 | 67.0 | 95.8 | 98.4 | 97.1 |
| | 50% DNC | 56.5 | 68.6 | 62.3 | 94.0 | 97.7 | 95.7 |
| CE | Inherent DNC | 61.7 | 76.2 | 68.6 | 96.3 | 98.8 | 97.5 |
| | 20% DNC | 59.6 | 74.0 | 66.3 | 95.9 | 98.6 | 97.2 |
| | 50% DNC | 54.4 | 68.5 | 61.1 | 92.5 | 97.0 | 94.5 |

## G.10 MORE ANALYTIC STUDY OF THE RELIABILITY

Here, we explore alternative strategies that dynamically manipulate the weights assigned to uncertainty $u_i$ and consensus $c_I$ in Eq. 1, *i.e.*,

$$w_i = (1 - u_i)\gamma + c_i(1 - \gamma) \tag{31}$$

where $\gamma$ is the balance parameter. From the results in Table 15, one could have the following conclusions: i) employ either uncertainty ($\gamma = 1$) or consensus ($\gamma = 0$) is inadequate in identifying DNC, leading to degraded performance; ii) thanks to the normalization in Eq. 2 and Eq. 5, a simple linear combination of uncertainty and consensus could yield considerable performance improvements. The results above indicate that our design for combining uncertainty and consensus is reasonable and effective.

Table 15: Analytic Experiments of the weighting strategy on the ICEWS-WIKI dataset under the "Non-name" setting.

| $\gamma$ | H@1 | H@5 | MRR |
|---|---|---|---|
| 0 | 51.8 | 62.4 | 57.0 |
| 0.2 | 53.4 | 64.4 | 58.7 |
| 0.4 | 54.9 | 66.5 | 60.4 |
| 0.5 | **56.5** | **68.6** | **62.3** |
| 0.6 | 55.1 | 67.1 | 60.8 |
| 0.8 | 53.9 | 66.6 | 60.0 |
| 1.0 | 52.5 | 65.7 | 58.8 |

## G.11 MORE EXPERIMENTS ON VARIOUS BACKBONES

The proposed RULE is a model-agnostic method, which could be employed in any mainstream vision-language backbones. To verify this, we further verify the effectiveness of RULE in various backbones. Specifically, we adopt SigLIP (Zhai et al., 2023) and BLIP (Li et al., 2022) as the backbones for extracting image and text features. Notably, we do not employ MLLM-based reasoning in the experiments for fairness and select the best-performing baselines from Tables 1-2 for comparisons. The results in Table 16 demonstrate that RULE outperforms all baselines across various backbones, which confirms the generality and effectiveness of RULE.

Table 16: Experiment results on the ICEWS-WIKI dataset with SigLIP and BLIP as backbone.

| Setting | Methods | SigLIP | | BLIP | |
|---|---|---|---|---|---|
| | | Non-name | All-attributes | Non-name | All-attributes |
| Inherent DNC | MEAformer | 45.3 | 93.8 | 33.4 | 95.1 |
| | UMAEA | 41.5 | 92.0 | 31.5 | 93.6 |
| | PMF | 45.2 | 92.3 | 30.8 | 95.3 |
| | Ours | **55.2** | **95.2** | **39.1** | **96.8** |
| 20% DNC | MEAformer | 42.2 | 89.0 | 28.9 | 95.0 |
| | UMAEA | 39.4 | 89.3 | 26.8 | 91.3 |
| | PMF | 37.8 | 89.0 | 26.6 | 89.7 |
| | Ours | **52.3** | **94.5** | **34.9** | **96.7** |
| 50% DNC | MEAformer | 32.2 | 88.5 | 19.6 | 93.2 |
| | UMAEA | 27.4 | 85.5 | 15.3 | 89.5 |
| | PMF | 28.1 | 83.9 | 16.8 | 80.9 |
| | Ours | **45.4** | **93.2** | **24.4** | **97.1** |

### G.12 MORE EXPERIMENTS ON FB15K-DB15K AND FB15K-YAGO15K BENCHMARKS

Here, we conduct additional experiments on the FB15K-DB15K and FB15K-YAGO15 benchmarks with their inherent handcrafted features. Notably, the raw data for these two datasets is no longer available, so we don't employ MLLM for reasoning at test-time. Here, we select the best-performing baselines from Tables 1-2 for comparisons. The results in Table 17-18 demonstrate that RULE outperforms all baselines on FB15K-DB15K and FB15K-YAGO15K datasets under the settings with various DNC ratios.

Table 17: Experiment results on the FB15K-DB15K dataset under the "Non-name" setting. "Inherent DNC" refers to the setting without any additional injected noise.

| Setting | Methods | H@1 | H@5 | MRR |
|---|---|---|---|---|
| Inherent DNC | MEAformer | 40.8 | 62.7 | 51.0 |
| | UMAEA | 40.8 | 62.0 | 50.7 |
| | PMF | 43.0 | 65.3 | 53.2 |
| | Ours | **44.4** | **63.6** | **53.8** |
| 20% DNC | MEAformer | 28.1 | 49.7 | 39.6 |
| | UMAEA | 29.5 | 51.4 | 41.1 |
| | PMF | 29.6 | 51.4 | 41.2 |
| | Ours | **36.0** | **56.0** | **45.9** |
| 50% DNC | MEAformer | 12.0 | 26.6 | 20.7 |
| | UMAEA | 15.0 | 32.1 | 24.7 |
| | PMF | 15.1 | 31.7 | 24.6 |
| | Ours | **20.3** | **36.9** | **28.9** |

Table 18: Experiment results on the FB15K-YAGO15K dataset under the "Non-name" setting.

| Setting | Methods | H@1 | H@5 | MRR |
|---|---|---|---|---|
| Inherent DNC | MEAformer | 31.8 | 50.5 | 40.7 |
| | UMAEA | 31.2 | 50.9 | 40.3 |
| | PMF | 34.6 | 55.0 | 44.3 |
| | Ours | **38.9** | **55.7** | **47.0** |
| 20% DNC | MEAformer | 20.5 | 38.1 | 30.1 |
| | UMAEA | 22.6 | 40.0 | 31.8 |
| | PMF | 23.7 | 42.0 | 33.2 |
| | Ours | **31.7** | **49.0** | **40.4** |
| 50% DNC | MEAformer | 10.3 | 20.7 | 16.2 |
| | UMAEA | 11.9 | 24.0 | 18.6 |
| | PMF | 11.3 | 22.8 | 17.9 |
| | Ours | **17.9** | **31.9** | **25.1** |

### G.13    Discussions with Active Learning Paradigm

One intuitive approach to address DNC is to introduce additional expert annotations and the active learning paradigm (Brame, 2016) might be the most representative paradigm. However, we argue that active learning is inadequate for addressing DNC for the following reasons:

- Existing MMEA-oriented active learning methods (Xu et al., 2024) focus on annotating E-E correspondences, which have not explored the establishment of E-A and A-A associations. As a result, even with expert involvement, active learning is unable to solve E-A or A-A NC. It is important to emphasize that these two types of NC are highly practical in real-world scenarios, with their total proportion exceeding 40% in commonly-used ICEWS benchmarks.

- Active learning requires expert involvement, which is time-consuming and labor-intensive. In contrast, RULE provides a robust learning paradigm in an automated manner. Moreover, even after careful manual annotation, over 50% of the data suffer from DNC challenge in real-world datasets, which indicates that expert annotation remains prone to errors.

- Active learning selects the most valuable samples for human annotation to improve model performance. However, most active learning methods require an initial set of labeled data for value model training before conducting active learning. As a result, the DNC challenge is still inevitable in the initial set, thus degrading the performance of the value model and then undermining the active learning.

### G.14    Discussions with Related Works

Recently, several related studies have explored to address noise issues in MMEA task. Among them, DESAlign (Wang et al., 2024) might be the most relevant to our work. However, DESAlign and the proposed RULE differ in the following aspects: i) Different Motivations. DESAlign focuses on addressing disparities in attribute counts or the absence of certain modalities, while RULE aims to mitigate the negative impacts of E-E NC, E-A NC, and A-A NC on intra-entity attribute fusion and inter-graph discrepancy elimination; ii) Different Key Ideas. DESAlign employs a robust semantic learning to mitigate semantic inconsistency and interpolates missing modality through a propagation strategy. In contrast, RULE not only adopts an intra-entity attribute fusion and an inter-graph loss to achieve robustness against DNC, but also leverages the test-time reasoning module to mine underlying associations between attributes and thus guarantees more accurate equivalent entity identification.

### G.15    Discussions about the Reliability Estimation

To alleviate the negative impact of DNC, it is desirable to establish a unified reliability principle so that both intra-entity and inter-graph NC could be identified. An intuitive approach is to estimate the reliability of inter-graph pairs by leveraging intra-entity pairs, as intra-entity NC may further contaminate inter-graph attribute pairs. However, such a vanilla internal-to-external approach is limited, since some mismatched entity pairs may result from annotation errors (as shown in Fig. 1 (a)) rather than intra-entity NC. To remedy this, we explore a novel external-to-internal estimation paradigm. On the one hand, the reliability of inter-graph correspondences could be directly assessed. On the other hand, since intra-entity NC would inevitably lead to mismatched inter-graph attribute pairs, these attribute pairs could be employed to estimate the reliability of intra-entity associations.

## H  CASE STUDY OF DUAL-LEVEL NOISY CORRESPONDENCE

In this section, we showcase some DNC examples from the ICEWS benchmark in Fig. 10.

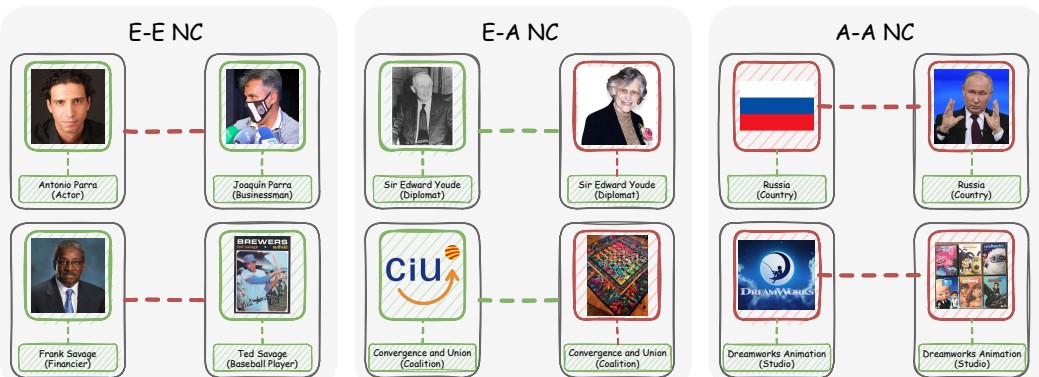

Figure 10: Examples of DNC in the ICEWS benchmark.

## I  CASE STUDY OF TEST-TIME CORRESPONDENCE REASONING

In this section, we visualize the test-time reasoning process in TTR module. Specifically, we present several representative successful and failure cases, and further provide visualizations of performing TTR on ambiguous and missing attributes.

**Successful Cases:** As illustrated in Fig. 11-12, the TTR module uncovers the underlying connections between image and name attribute pairs, thus boosting performance.

**Failure Cases:** Although the TTR module could rectify MMEA results by mining implicit associations, we still observe several failure cases as illustrated in Fig. 13. According to the visualization results, we attribute these failure cases to the following two reasons. On the one hand, the MLLM might lack enough knowledge to perform correct reasoning. For example, when the query refers to the abbreviation of Toronto Metropolitan University, the MLLM does not possess the necessary background knowledge and thus cannot find the corresponding university logo. On the other hand, the MLLM might sometimes resort to shortcut cues rather than performing deep reasoning, leading to undiscovered associations even after reasoning. For instance, the MLLM may still rely on visual similarity when identifying coastal cities, while ignoring underlying semantic cues that indicate the corresponding country.

**Ambiguous Cases:** TTR could perform correct test-time reasoning even when the attributes are ambiguous. As illustrated in Fig. 14, the image attributes of the candidates are highly similar. Nevertheless, the TTR module is able to uncover meaningful associations, such as identifying the same individual across different scenes or recognizing variations in clothing and appearance.

**Missing Cases:** Multimodal knowledge graphs often contain missing attributes, particularly missing images (Chen et al., 2023b). We present several visualizations in Fig. 15 to illustrate how TTR module behaves under missing attributes scenario. From the results, we observe that when the image attribute of the correct candidate is missing, the TTR module assigns low similarity scores to other images and instead relies on the available and more reliable textual attributes for more accurate reasoning. As a result, the TTR module could still achieve correct alignment even in the presence of missing attributes, demonstrating the effectiveness and robustness of RULE.

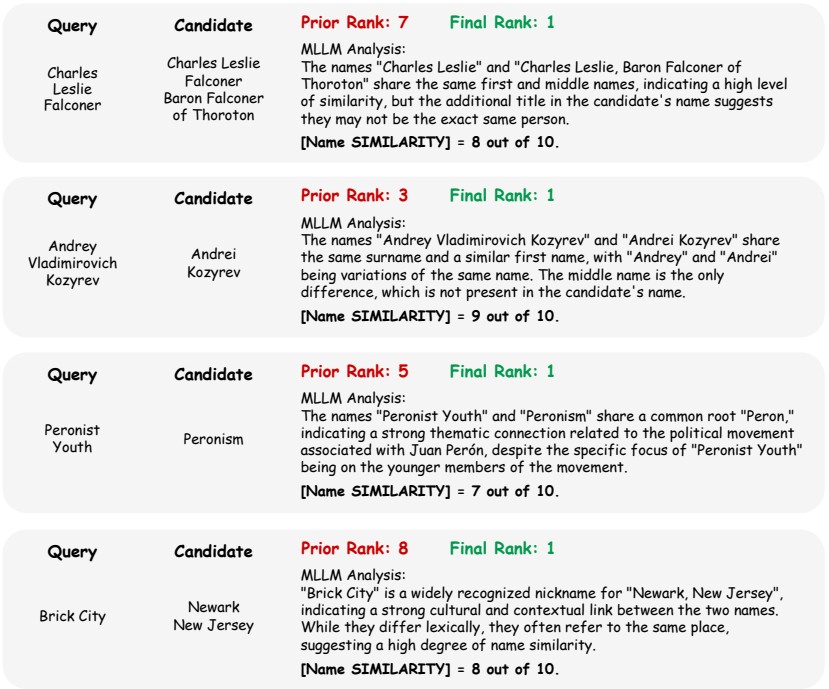

Figure 11: Reasoning process for the name attributes in test-time reasoning.

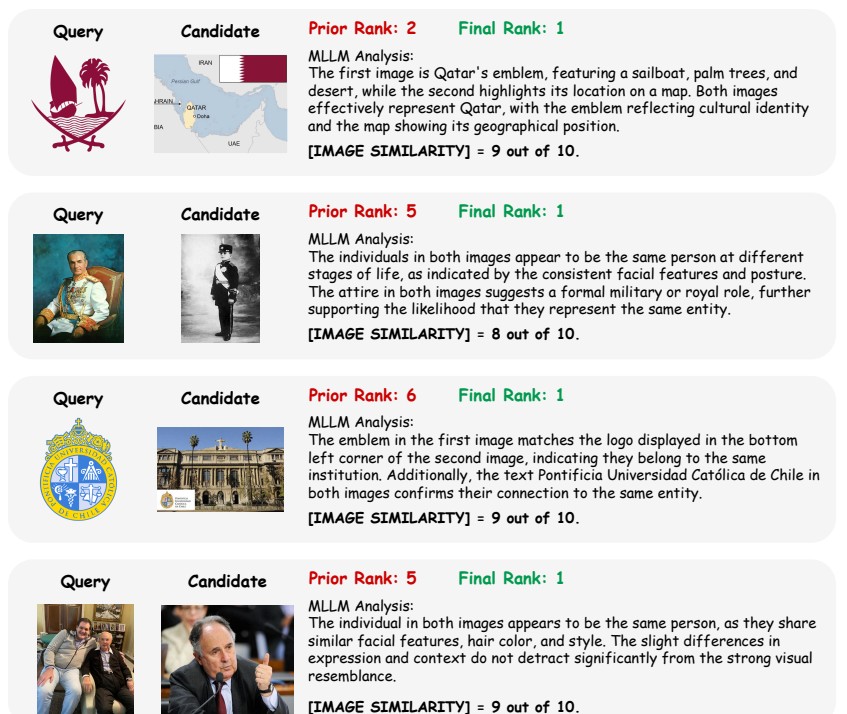

Figure 12: Reasoning process for the image attributes in test-time reasoning.

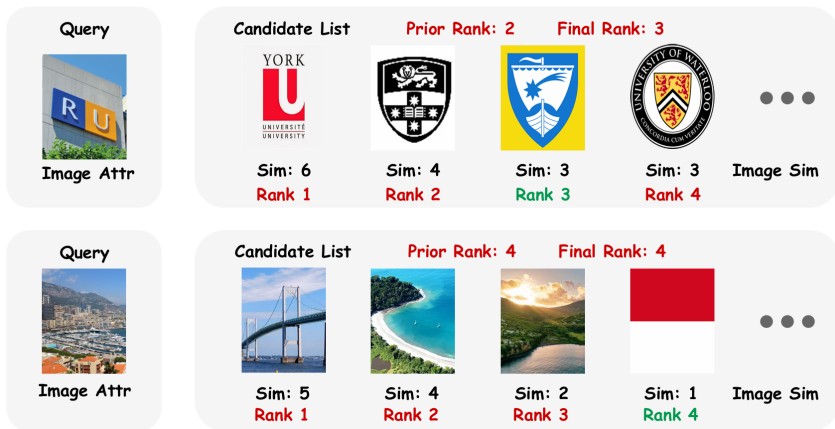

Figure 13: Failure cases in test-time reasoning.

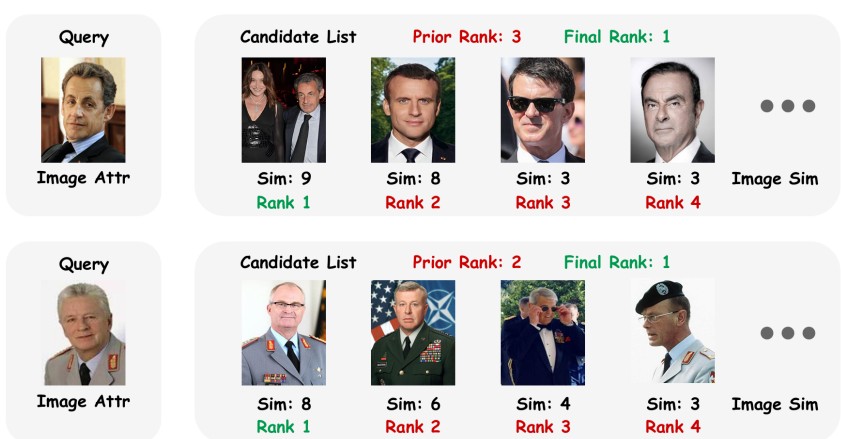

Figure 14: Ambiguous attributes in test-time reasoning.

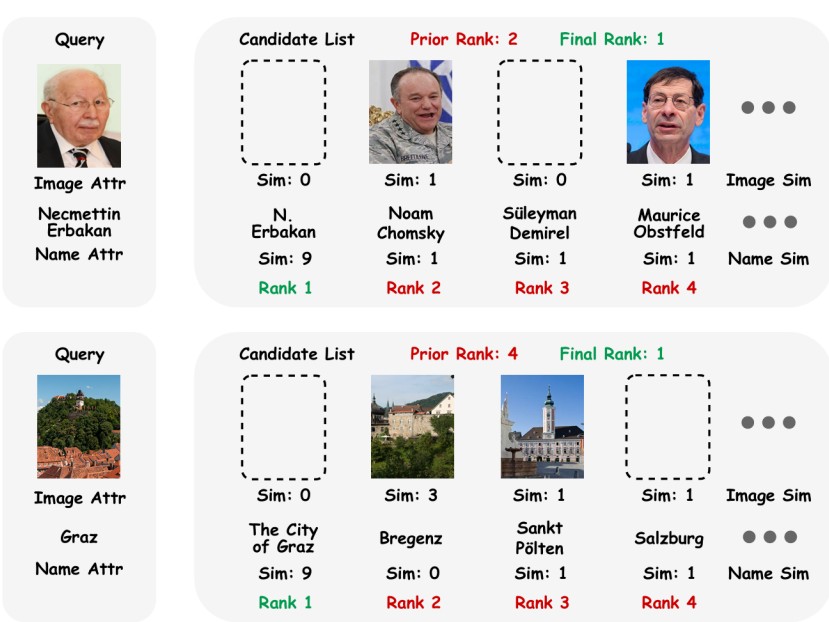

Figure 15: Missing attributes in test-time reasoning.

