# OpenReview forum: "Learning with Dual-level Noisy Correspondence for Multi-modal Entity Alignment"
_ICLR.cc/2026/Conference — ICLR 2026 Oral_

### Official Review · Reviewer_1ndq · 2025-10-29

**Soundness:** 4
**Presentation:** 3
**Contribution:** 4
**Rating:** 8
**Confidence:** 5

**Summary:**

In this paper, the authors study a new and practical problem in MMEA called dual-level noisy correspondence (DNC), where both intra-entity and inter-graph correspondences might be noisy. To address this, the authors propose a robust framework (RULE) that estimates correspondence reliability based on a two-fold principle. With the estimated reliabilities, RULE performs noise-aware attribute fusion and discrepancy elimination, and further introduces a correspondence reasoning module to enhance test-time robustness. The authors conduct thoughtful experiment designs and the results show that RULE achieves significant improvements over existing methods. Overall evaluation, this is a good paper with interesting motivation and solid results.

**Strengths:**

This paper reveals a highly practical yet underexplored challenge (DNC) in MMEA. I appreciate the motivation of this work and believe that tackling DNC is valuable, as intra-entity and inter-graph noisy correspondences are often inevitable in real-world datasets. Particularly, the authors further provide statistical analysis of the noise distribution in real-world datasets in the Appendix, which is convincing and further highlights the necessity of addressing DNC. To address the DNC problem, the paper proposes a unified principle to estimate the reliability of correspondences across different levels. Based on the estimated reliability, RULE enhances robustness against DNC in both the training and inference phases, pioneering a full-process robust learning paradigm for MMEA. Overall, the paper tackles a meaningful and practical problem with a technically sound solution.

**Weaknesses:**

Although the paper is well-motivated and presents a technical-sound solution, I have several questions and suggestions for further improvement as follows:
- There are some prior efforts on learning with noisy correspondence (NC) in cross-modal retrieval. Since DNC could be viewed as a special case of NC in MMEA, I wonder whether existing NC-oriented methods are adequate for addressing the DNC challenge. I encourage the authors to clarify the distinctions and advantages of RULE over prior NC-oriented studies. Moreover, I notice that the paper ``Tackling Uncertain Correspondences for Multi-Modal Entity Alignment [A]” share some similar motivation with this submission, could the authors give some discussions?
- I am curious about how the DNC ratios (0%, 20%, and 50%) used in the experimental settings are determined. Are these noise ratios chosen based on empirical observations or following prior studies? It would be helpful if the authors could clarify the reasons behind these choices.
- As far as I know, uncertainty has been widely used as a principle for identifying noise input in classification tasks. However, I’m not clear about why uncertainty is insufficient to measure the reliabilities of the correspondences in MMEA. In other words, what motivates the introduction of the consensus principle?
- I am particularly curious about how the TTR module rectifies correspondences during test time. It is recommended that the authors include visualization examples to illustrate this process.

**Questions:**

- Provide more discussion on the differences between the proposed method and existing NC-oriented approaches and the related work [A].
- Explain the choice of noise ratios (0%, 20%, 50%) in the experiments.
- Justify the necessity of the consensus principle.
- Visualize the test-time reasoning process to offer more intuitive insights into how the TTR module rectifies correspondences.

---

> ### Author Response · Authors · 2025-11-17
>
> Thanks for the insightful reviews. We will answer your questions one by one in the following.
>
> > Q1: There are some prior efforts on learning with noisy correspondence (NC) in cross-modal retrieval. Since DNC could be viewed as a special case of NC in MMEA, I wonder whether existing NC-oriented methods are adequate for addressing the DNC challenge. I encourage the authors to clarify the distinctions and advantages of RULE over prior NC-oriented studies. Moreover, I notice that the paper ``Tackling Uncertain Correspondences for Multi-Modal Entity Alignment [A]” share some similar motivation with this submission, could the authors give some discussions?
>
> **A1**: Thanks for your comments. To the best of our knowledge, **such a Dual-level Noisy Correspondence (DNC) challenge has not been explored so far**, and existing NC-oriented methods are intractable to solve it for the following reasons: i) since most of them are tailored to identify NC at the external level (e.g., cross-modal and cross-graph scenarios), they overlook that NC would also exist within the internal graph, not to mention the more complex multi-level NC revealed in our work; ii) existing methods focus on achieving robust discrepancy elimination at a specific level (e.g., image-text alignment), while DNC spans multiple levels and would mislead both the multi-modal fusion and cross-graph alignment; iii) although existing NC-oriented methods could achieve robustness during the training stage, RULE further enhance robustness at test time, which has yet to be explored in prior work.
>
> As for the mentioned TMEA [A], we argue that TMEA and the proposed RULE differ in the following aspects:
> i) **Different Motivations.** TMEA aims to address weak associations and missing modalities in MMEA, whereas RULE focuses on handling incorrect inter-entity and inter-graph correspondence;
> ii) **Different Key Ideas.** TMEA develops an inter-modal commonality enhancement mechanism to address weak semantic associations between modalities, while RULE employs robust intra-entity attribute fusion and inter-graph discrepancy elimination to mitigate the negative impact of DNC. In addition, TMEA focus on achieving robustness against noise during the training stage. In contrast, RULE not only employs the robust attribute fusion module to further enhance test-time robustness against NC, but also proposes a reasoning module to mine underlying associations between attributes.
>
> > Q2: I am curious about how the DNC ratios (0%, 20%, and 50%) used in the experimental settings are determined. Are these noise ratios chosen based on empirical observations or following prior studies? It would be helpful if the authors could clarify the reasons behind these choices.
>
> **A2**: Thanks for your comments. The designs about the noise ratios of 0%, 20%, and 50% **follow the common practice in prior works on noisy labels [B] and noisy correspondences [C]**, which are employed to comprehensively evaluate the robustness of RULE under extreme noise scenarios.
>
> Note that even without artificially injected noise, **the Inherent DNC setting in Tables 1-2 already suffers from a severe DNC challenge**, as real-world MMEA datasets inherently contain DNC cases. As reported in Appendix B, even in carefully annotated datasets such as ICEWS-WIKI and ICEWS-YAGO, more than 50% of entity pairs suffer from DNC, which further supports that DNC is a highly practical and pressing issue in real-world scenarios.

---

> > ### Author Response · Authors · 2025-11-17
> >
> > > Q3: As far as I know, uncertainty has been widely used as a principle for identifying noise input in classification tasks. However, I’m not clear about why uncertainty is insufficient to measure the reliabilities of the correspondences in MMEA. In other words, what motivates the introduction of the consensus principle?
> >
> > **A3**: Thanks for your comments. In the submission, we argue that although the formulated uncertainty would help to identify noisy correspondence, **a low uncertainty does not necessarily indicate a correct correspondence**. According to Theorem 1 in Section 2.2.2 and its proof in Appendix E, **we conclude that a low uncertainty does not guarantee that the highest belief is assigned to the correct correspondence**. Intuitively, evidence vectors such as [0,0,0,1] and [0,1,0,0] yield the same uncertainty. However, only one of them might indicate the correct correspondence, i.e., the second or the fourth candidate is actually correct.
> >
> > To remedy this, we propose a novel consensus principle that leverages ground-truth annotations to further estimate the reliability of correspondence. Moreover, since annotations are unavailable during inference, we propose to estimate the correct correspondence via a greedy strategy based on marginal contribution, as described in Section 2.2.2.
> >
> > > Q4: I am particularly curious about how the TTR module rectifies correspondences during test time. It is recommended that the authors include visualization examples to illustrate this process.
> >
> > **A4**: Thanks for your valuable suggestions. We have provided several visualization examples in Appendix F.5 and Appendix I. According to these results, we observe that the TTR module could mine the underlying associations between attributes. For instance, different textual expressions of the same entity (e.g., Brick City and Newark, New Jersey), different images of the same entity (e.g., the Qatar national flag and a map of Qatar), and portraits of the same person at different ages (e.g., a general in youth vs. in old age). In other words, **the MLLM could uncover seemingly dissimilar but inherently identical attributes, which is often overlooked by existing MMEA methods**. Accordingly, we could perform a robust fusion of the test-time reasoning results and the prior results, thus yielding rectified MMEA results.
> >
> > [A] Tackling Uncertain Correspondences for Multi-Modal Entity Alignment, NeurIPS, 2024.
> >
> > [B] Learning with Noisy Correspondence for Cross-modal Matching, NeurIPS, 2021.
> >
> > [C] Learning with noisy labels, NeurIPS, 2013.

---

### Official Review · Reviewer_ny9k · 2025-10-29

**Soundness:** 3
**Presentation:** 4
**Contribution:** 4
**Rating:** 8
**Confidence:** 5

**Summary:**

This paper makes the first attempt to explore dual-level noisy correspondences (DNC) in the field of multi-modal entity alignment (MMEA). To address this issue, the proposed RULE could not only achieve robust cross-graph learning and multi-modal fusion during training, but also incorporates a correspondence reasoning module to enhance test-time robustness and thus achieves more accurate entity alignment across heterogeneous multi-modal knowledge graphs. Extensive experiments demonstrate that RULE outperforms state-of-the-art MMEA methods, while showing significantly slower performance degradation as noise levels increase.

**Strengths:**

1. This paper is well-motivated and is clearly organized.
2. The intuitive illustrations in Figure 1 make the DNC phenomenon and its negative impacts clear.
3. The experiments are convincing, e.g., the visualizations (Figure 3b) of the reliability estimation, pair division process (Figure 4) further prove the effectiveness of the proposed RULE.
4. Most impressively, Appendix B not only analyzes the noise statistics in real-world datasets but also discusses the underlying causes of DNC. Correspondingly, the framework demonstrates strong performance under both noise-injected and inherent DNC settings, further validating the presence of DNC in real-world datasets.

**Weaknesses:**

1. Since both uncertainty and consensus are estimated through cross-graph relationships, I don’t understand why such relationships could be employed to estimate reliability of intra-graph attributes. A more intuitive and detailed explanation of this design would be very helpful for understanding.
2. In my understanding, the MLLM-based reasoning module aims to uncover underlying associations between attributes. I am interested in what kinds of associations can be further mined by the MLLM, for example, beyond the cases illustrated in Figure 1(c). Furthermore, I wonder whether the proposed MLLM-based reasoning module could be extended to other tasks, such as image-text retrieval.
3. Since the proposed RULE is a model-agnostic framework, which means RULE could be applied on various backbones. It would further strengthen the contribution if the authors could demonstrate the generalizability of RULE across more diverse backbones.
4. I want to see more details about the computational complexity of the proposed RULE. In particular, the MLLM-based reasoning module may introduce considerable time and memory. It would be valuable for the authors to provide a detailed analysis of the reasoning time, GPU requirements.

**Questions:**

Please see the Weaknesses.

---

> ### Author Response · Authors · 2025-11-17
>
> Thanks for the insightful reviews. We will answer your questions one by one in the following.
>
> > Q1: Since both uncertainty and consensus are estimated through cross-graph relationships, I don’t understand why such relationships could be employed to estimate reliability of intra-graph attributes. A more intuitive and detailed explanation of this design would be very helpful for understanding.
>
> **A1**: Thanks for your comments. To alleviate the negative impact of DNC, it is desirable to establish a unified reliability principle so that both intra-entity and inter-graph NC could be identified.
> An intuitive approach is to estimate the reliability of inter-graph pairs by leveraging intra-entity pairs, as intra-entity NC may further contaminate inter-graph attribute pairs (as discussed in Appendix B.2).
> However, such a vanilla internal-to-external approach is limited, since some mismatched entity pairs may result from annotation errors (as shown in Fig. 1 (a)) rather than intra-entity NC.
> To remedy this, **we explore a novel external-to-internal estimation paradigm**.
> On the one hand, the reliability of inter-graph correspondences could be directly assessed.
> On the other hand, since intra-entity NC would inevitably lead to mismatched inter-graph attribute pairs, these attribute pairs could be employed to estimate the reliability of intra-entity associations.
>
> **In the revised manuscript, we have supplemented the detailed discussions about the reliability estimation paradigm in Appendix G.15**.
>
> > Q2: In my understanding, the MLLM-based reasoning module aims to uncover underlying associations between attributes. I am interested in what kinds of associations can be further mined by the MLLM, for example, beyond the cases illustrated in Figure 1(c). Furthermore, I wonder whether the proposed MLLM-based reasoning module could be extended to other tasks, such as image-text retrieval.
>
> **A2**: Thanks for your constructive comments. We have provided several visualization results of the mined underlying attribute associations in Appendix I (Figures 11 and 12). Specifically, the MLLM-based reasoning module could i**dentify different expressions referring to the same entity** (e.g., Brick City and Newark, New Jersey), **different images of the same entity** (e.g., the Qatar national flag and a map of Qatar), and **different temporal portraits of the same individual** (e.g., a general in youth vs. in old age).
>
> In addition, we believe that the proposed MLLM-based reasoning module is potential for applications in image–text retrieval, where it could uncover seemingly dissimilar but inherently identical image-text pairs. For example, the text query “Cristiano Ronaldo” and an image showing the Portugal national football team jersey, the text description “Donald Trump” and an image of the U.S. national flag.
>
> > Q3: Since the proposed RULE is a model-agnostic framework, which means RULE could be applied on various backbones. It would further strengthen the contribution if the authors could demonstrate the generalizability of RULE across more diverse backbones.
>
> **A3**: In order to address your concerns, **we further verify the effectiveness of RULE in various backbones**. Specifically, we adopt SigLIP and BLIP as the backbones for extracting image and text features. For fair comparisons, we do not employ MLLM-based reasoning in the experiments. For your convenience, we attach the corresponding results in the following tables.

---

> > ### Author Response · Authors · 2025-11-17
> >
> > Table A: **Experiment Results** on the ICEWS-WIKI dataset with SigLIP and BLIP as backbones.
> >
> > | Setting      | Methods   | SigLIP Non-name H@1 | SigLIP All-attributes H@1 | BLIP Non-name H@1 | BLIP All-attributes H@1 |
> > | ------------ | --------- | ------------------- | ------------------------- | ----------------- | ----------------------- |
> > | Inherent DNC | MEAformer | 45.3                | 93.8                      | 33.4              | 95.1                    |
> > |              | UMAEA     | 41.5                | 92.0                      | 31.5              | 93.6                    |
> > |              | PMF       | 45.2                | 92.3                      | 30.8              | 95.3                    |
> > |              | Ours      | **55.2**            | **95.2**                  | **39.1**          | **96.8**                |
> > | 20% DNC      | MEAformer | 42.2                | 89.0                      | 28.9              | 95.0                    |
> > |              | UMAEA     | 39.4                | 89.3                      | 26.8              | 91.3                    |
> > |              | PMF       | 37.8                | 89.0                      | 26.6              | 89.7                    |
> > |              | Ours      | **52.3**            | **94.5**                  | **34.9**          | **96.7**                |
> > | 50% DNC      | MEAformer | 32.2                | 88.5                      | 19.6              | 93.2                    |
> > |              | UMAEA     | 27.4                | 85.5                      | 15.3              | 89.5                    |
> > |              | PMF       | 28.1                | 83.9                      | 16.8              | 80.9                    |
> > |              | Ours      | **45.4**            | **93.2**                  | **24.4**          | **97.1**                |
> >
> > The results demonstrate that RULE outperforms all baselines across various backbones, which confirms the generality and effectiveness of RULE.
> >
> > > Q4: I want to see more details about the computational complexity of the proposed RULE. In particular, the MLLM-based reasoning module may introduce considerable time and memory. It would be valuable for the authors to provide a detailed analysis of the reasoning time, GPU requirements.
> >
> > **A4**: Thanks for your suggestions. In this paper, we have conducted an analysis of computational complexity in Appendix G.8. Specifically, **we analyze the reasoning time and memory cost of the MLLM-based reasoning module and present their performance in Table 13**.** For your convenience, we attach the corresponding analysis results in the following tables. The time unit is seconds. "w/o TTR" refers to employing RULE during the training stage, without using the TTR.
> >
> > Table 13: Experiment results for the complexity analysis on the ICEWS-WIKI dataset.
> >
> > | Methods        | H@1 (Non-name) | Time Cost (Non-name) | H@1 (All-attributes) | Time Cost (All-attributes) | Memory Consumption  |
> > | -------------- | -------------- | -------------------- | -------------------- | -------------------------- | ------------------- |
> > | w/o TTR        | 56.5           | 103                  | 94.0                 | 109                        | 1GPU $\times$ ~16GB |
> > | Qwen2.5-VL 3B  | 57.2           | 2122                 | 96.5                 | 1008                       | 1GPU $\times$ ~8GB  |
> > | Qwen2.5-VL 7B  | 57.4           | 2690                 | 97.1                 | 1329                       | 1GPU $\times$ ~8GB  |
> > | Qwen2.5-VL 72B | 58.2           | 10043                | 97.7                 | 4373                       | 8GPU $\times$ ~20GB |
> >
> > From the results, one can observe that **employing Qwen2.5-VL 3B or 7B for TTR leads to a considerable performance boost, with up to a 5× speedup**. Furthermore, when resources are limited, RULE can be deployed without MLLMs. **Even without TTR, RULE still significantly outperforms the strongest baselines**, achieving 56.5% vs. 43.9% (best-performing baseline) in the ''Non-name'' setting and 94.0% vs. 91.9% (best-performing baseline) in the ''All-attributes'' setting. Note that we introduce test-time correspondence reasoning inspired by **test-time scaling** [C], thus further enhancing the performance.
> >
> > [A] MEAformer: Multi-modal Entity Alignment Transformer for Meta Modality Hybrid, MM, 2023.
> >
> > [B] Progressively Modality Freezing for Multi-Modal Entity Alignment, ACL, 2024.
> >
> > [C] s1: Simple test-time scaling, arXiv, 2025.

---

### Official Review · Reviewer_Xav9 · 2025-10-29

**Soundness:** 3
**Presentation:** 3
**Contribution:** 3
**Rating:** 6
**Confidence:** 4

**Summary:**

The paper addresses a critical issue in MMEA, namely Dual-level Noisy Correspondence (DNC), which refers to the misalignments at both the intra-entity, entity-attribute, and inter-graph, entity-entity, attribute-attribute, levels. This noise can severely degrade performance in multi-modal knowledge graph alignment tasks, which is a common problem in real-world benchmarks. To tackle this issue, the authors propose a framework called RULE, which robustly estimates the reliability of these correspondences using a two-fold principle, uncertainty and consensus. RULE mitigates the impact of noisy correspondences during training and incorporates a test-time correspondence reasoning module to enhance the robustness during inference. Experiment demonstrates that RULE outperforms existing state-of-the-art methods across several datasets and noise settings.

**Strengths:**

1. The identification and study of dual-level noisy correspondence is a novel and significant contribution to the MMEA task. It acknowledges that real-world knowledge graphs often contain significant noise both within entities and across graphs, which previous methods have largely ignored.

2. The proposed RULE framework leverages reliability estimation using uncertainty and consensus to combat noisy correspondences. The test-time correspondence reasoning module is a key innovation that improves entity identification at inference time, ensuring better alignment even when certain correspondences are noisy or unreliable.

**Weaknesses:**

1. This paper includes some parameter analysis like trade-off parameter λ, threshold β, and temperature τ, but there is no comprehensive discussion of how these hyperparameters affect the model’s robustness under various noise settings. For instance, how does RULE perform with different datasets or in cases where the noise level is very high, e.g., >50%?

2. RULE relies heavily on pre-trained CLIP models for image and text embeddings. While this is a reasonable approach, it could limit the flexibility of the model in scenarios where domain-specific or fine-grained embeddings are needed.

3. It seems lack of how the framework handles extreme cases, such as when the noise level exceeds 50%, or when the correspondence noise is extremely high in one modality (e.g., image attributes).

4. While the TTR module improves the robustness during inference, it may introduce additional computational overhead. The reasoning process involves complex reasoning steps and large language model queries, which could be slow and computationally expensive, especially for large-scale graphs.

**Questions:**

1. Could you provide a more detailed analysis of how the hyperparameters (λ, β, τ) affect the performance of RULE under different noise levels? Are there specific configurations where the model is more or less sensitive? It would be useful to explore whether hyperparameter tuning plays a significant role in noise resilience.

2. How do you think RULE compares to methods like graph matching or other representation learning-based approaches that also handle noisy correspondences? Some methods like [1] also addresses noise in multi-modal entity alignment, which names semantic consistency. It would be better if you can compare and analyze them.

3. I`m wondering how does RULE perform in extreme noise scenarios like >50% noise or heavily corrupted attributes in one modality? Would the model still perform robustly, or would it need additional modifications?

---

> ### Author Response · Authors · 2025-11-17
>
> Thanks for your constructive reviews and suggestions. In the following, we will answer your questions one by one.
>
> > Q1: This paper includes some parameter analysis like trade-off parameter λ, threshold β, and temperature τ, but there is no comprehensive discussion of **how these hyperparameters affect the model’s robustness under various noise settings**. Are there specific configurations where the model is more or less sensitive? (Weakness 1 and Questions 1)
>
> **A1**: Thank you for your valuable comments. In the submission, we have conducted hyperparameter analyses regarding $\lambda$, $\beta$, and $\tau$ under the setting of 50% DNC ratio on the ICEWS-WIKI benchmark, and present the results in Appendix G.2 (Fig. 7).
>
> **In response to your suggestions, we carry out additional hyperparameter analyses under more noise settings.** In particular, we investigate the influence of three hyperparameters **under injected DNC ratios of 0%, 20%, 50%, and 70%**. The corresponding results are summarized in the following tables.
>
> Table A: Hyperparameters Analysis of the Trade-off Parameter $\lambda$ under various noise settings regarding the Hits@1 metric.
>
> | Trade-off Parameter<br />$\lambda$ | Non-name (Inherent DNC) | Non-name (20% DNC) | Non-name (50% DNC) | Non-name (70% DNC) | All-attributes (Inherent DNC) | All-attributes (20% DNC) | All-attributes (50% DNC) | All-attributes (70% DNC) |
> | ---------------------------------- | ----------------------- | ------------------ | ------------------ | ------------------ | ----------------------------- | ------------------------ | ------------------------ | ------------------------ |
> | 1e-2                               | 40.4                    | 35.5               | 31.5               | 26.5               | 44.2                          | 40.6                     | 37.9                     | 33.4                     |
> | 1e-3                               | 56.3                    | 54.2               | 52.4               | 41.2               | 94.2                          | 93.6                     | 93.2                     | 89.2                     |
> | 5e-4                               | 60.3                    | 58.4               | 55.7               | 46.5               | 95.6                          | 93.9                     | 93.2                     | 90.0                     |
> | 2e-4                               | 61.4                    | 58.9               | 56.1               | 47.1               | 96.3                          | 95.4                     | 93.5                     | 90.9                     |
> | 1e-4                               | **62.1**                | **60.4**           | **56.5**           | **48.2**           | **96.7**                      | **95.8**                 | **94.0**                 | **91.3**                 |
> | 5e-5                               | 61.8                    | 60.1               | 56.3               | 48.1               | 96.5                          | 95.3                     | 93.7                     | 91.0                     |
> | 2e-5                               | 60.9                    | 59.4               | 56.1               | 47.4               | 96.0                          | 94.8                     | 93.4                     | 90.2                     |
> | 1e-5                               | 60.1                    | 59.1               | 55.8               | 47.0               | 95.7                          | 94.2                     | 93.0                     | 89.7                     |

---

> > ### Author Response · Authors · 2025-11-17
> >
> > Table B: Hyperparameters Analysis of the Threshold $\beta$ under various noise settings regarding the Hits@1 metric.
> >
> > | Threshold<br />$\beta$ | Non-name (Inherent DNC) | Non-name (20% DNC) | Non-name (50% DNC) | Non-name (70% DNC) | All-attributes (Inherent DNC) | All-attributes (20% DNC) | All-attributes (50% DNC) | All-attributes (70% DNC) |
> > | ---------------------- | ----------------------- | ------------------ | ------------------ | ------------------ | ----------------------------- | ------------------------ | ------------------------ | ------------------------ |
> > | 0                      | 59.3                    | 56.8               | 55.7               | 38.9               | 94.5                          | 93.7                     | 93.3                     | 89.6                     |
> > | 0.1                    | 60.1                    | 58.1               | 55.9               | 41.2               | 95.7                          | 94.6                     | 93.4                     | 90.2                     |
> > | 0.2                    | 61.2                    | 58.4               | 56.2               | 46.5               | 96.3                          | 95.4                     | 93.5                     | 91.0                     |
> > | 0.3                    | **62.1**                | **60.4**           | **56.5**           | **48.2**           | **96.7**                      | **95.8**                 | **94.0**                 | **91.3**                 |
> > | 0.4                    | 61.4                    | 59.1               | 56.1               | 47.3               | 96.4                          | 95.7                     | 93.6                     | 90.7                     |
> > | 0.5                    | 59.6                    | 57.3               | 55.8               | 45.3               | 96.1                          | 94.9                     | 93.6                     | 90.3                     |
> > | 0.6                    | 57.4                    | 54.2               | 53.5               | 41.6               | 94.6                          | 93.8                     | 93.4                     | 89.6                     |
> > | 0.7                    | 48.9                    | 50.2               | 52.8               | 35.8               | 93.5                          | 92.7                     | 92.3                     | 88.9                     |
> >
> > Table C: Hyperparameters Analysis of the Temperature $\tau$ under various noise settings regarding the Hits@1 metric.
> >
> > | Temperature<br />$\tau$ | Non-name (Inherent DNC) | Non-name (20% DNC) | Non-name (50% DNC) | Non-name (70% DNC) | All-attributes (Inherent DNC) | All-attributes (20% DNC) | All-attributes (50% DNC) | All-attributes (70% DNC) |
> > | ----------------------- | ----------------------- | ------------------ | ------------------ | ------------------ | ----------------------------- | ------------------------ | ------------------------ | ------------------------ |
> > | 0.01                    | 43.1                    | 37.5               | 32.4               | 28.1               | 79.3                          | 75.2                     | 73.7                     | 65.6                     |
> > | 0.02                    | 50.2                    | 45.2               | 39.7               | 39.2               | 82.1                          | 78.9                     | 75.0                     | 70.3                     |
> > | 0.05                    | 61.2                    | 58.7               | 55.3               | 46.7               | 95.4                          | 94.9                     | 93.6                     | 90.1                     |
> > | 0.07                    | **62.1**                | **60.4**           | **56.5**           | **48.2**           | **96.7**                      | **95.8**                 | 94.0                     | **91.3**                 |
> > | 0.1                     | 60.8                    | 59.8               | 55.7               | 48.0               | 96.6                          | 95.6                     | **94.3**                 | 90.8                     |
> > | 0.2                     | 58.5                    | 56.6               | 54.7               | 46.3               | 95.1                          | 94.5                     | 92.4                     | 88.4                     |
> > | 0.5                     | 44.6                    | 37.4               | 31.1               | 29.4               | 90.2                          | 88.8                     | 85.4                     | 80.5                     |
> > | 1                       | 41.0                    | 36.0               | 31.2               | 25.3               | 82.4                          | 79.6                     | 77.3                     | 73.9                     |

---

> > > ### Author Response · Authors · 2025-11-17
> > >
> > > From the results, one could have the following conclusions: i) **RULE exhibits stable performance when $\lambda$ is within the range of $[2e-5, 5e-4]$, $\tau$ within $[0.05, 0.2]$, and $\beta$ within $[0.2, 0.4]$**;
> > > (ii) under various noise ratio settings, there exists a set of optimal hyperparameters, with the trade-off parameter $\lambda = 1\mathrm{e}{-4}$, threshold $\beta = 0.3$, and temperature $\tau = 0.07$.
> > >
> > > > Q2: RULE relies heavily on pre-trained CLIP models for image and text embeddings. While this is a reasonable approach, it could limit the flexibility of the model in scenarios where domain-specific or fine-grained embeddings are needed.
> > >
> > > **A2**: Thanks for the constructive comments. We would like to clarify that in our approach, the embeddings are first extracted using frozen pre-trained models such as CLIP, and subsequently adapted to different datasets through a trainable fully connected layer. In other words, our method **effectively handles scenarios that require domain-specific or fine-grained embeddings**. This embedding-learning paradigm is well established in the MMEA community and has been widely adopted in prior work, including PMF [A], MEAformer [B], UMAEA [C], and DESAlign [D].
> > >
> > > > Q3: It seems lack of how the framework handles extreme cases, such as when the noise level exceeds 50%, or when the correspondence noise is extremely high in one modality (e.g., image attributes). I`m wondering how does RULE perform in extreme noise scenarios like >50% noise or heavily corrupted attributes in one modality? Would the model still perform robustly, or would it need additional modifications? (Weakness 3 and Questions 3)
> > >
> > > **A3**: Thanks for your valuable suggestions. In the submission, we have carried out experiments in Fig. 3 (a) and Fig. 8 to verify the effectiveness of the proposed RULE under various DNC ratios, covering manually injected DNC noise ratios from 0.0 to 0.7.
> > >
> > > To further address your concerns, **we conduct additional experiments on the ICEWS-WIKI and DBP15K ZH-EN benchmarks in extreme noise scenarios**. Specifically, to comprehensively validate the robustness of RULE against extreme noise, **we adopt the DNC, E–E NC, E–A NC, and A–A NC settings for evaluations with noise ratios varying from 0.6 to 0.8**. Note that the E–A NC setting corresponds to the scenario where the correspondence noise is extremely high in one modality, while the A–A NC setting corresponds to the scenario that there are heavily corrupted attributes in one modality. The corresponding results (regarding Hits@1) are presented in the following tables. Notably, for fair comparisons, we do not employ MLLM-based reasoning in the experiments.

---

> ### Author Response · Authors · 2025-11-17
>
> Table D: Comparisons with state-of-the-art methods on Non-name setting under extreme noise scenarios.
>
> | Noise Ratios | Method    | ICEWS-WIKI: DNC | ICEWS-WIKI: E-E NC | ICEWS-WIKI: E-A NC | ICEWS-WIKI: A-A NC | DBP15K ZH-EN: DNC | DBP15K ZH-EN: E-E NC | DBP15K ZH-EN: E-A NC | DBP15K ZH-EN: A-A NC |
> | ------------ | --------- | --------------- | ------------------ | ------------------ | ------------------ | ----------------- | -------------------- | -------------------- | -------------------- |
> | 60%          | MCLEA     | 20.5            | 24.0               | 32.9               | 43.3               | 50.1              | 52.5                 | 72.1                 | 74.2                 |
> |              | MEAformer | 38.2            | 44.2               | 54.2               | 54.4               | 59.3              | 65.0                 | 73.4                 | 81.1                 |
> |              | UMAEA     | 32.5            | 33.4               | 37.6               | 48.6               | 56.5              | 62.5                 | 70.3                 | 77.1                 |
> |              | PMF       | 34.6            | 36.7               | 42.7               | 52.9               | 62.1              | 64.4                 | 72.3                 | 80.2                 |
> |              | RULE      | **54.9**        | **59.8**           | **59.0**           | **61.9**           | **69.9**          | **72.8**             | **79.4**             | **85.1**             |
> | 70%          | MCLEA     | 17.1            | 20.8               | 30.7               | 42.5               | 42.9              | 46.0                 | 69.8                 | 73.1                 |
> |              | MEAformer | 34.7            | 40.2               | 52.7               | 53.7               | 45.2              | 57.3                 | 71.4                 | 80.0                 |
> |              | UMAEA     | 29.2            | 31.8               | 35.7               | 46.8               | 43.9              | 56.4                 | 67.9                 | 76.0                 |
> |              | PMF       | 32.3            | 34.1               | 40.9               | 51.4               | 56.3              | 59.9                 | 70.6                 | 79.5                 |
> |              | RULE      | **48.2**        | **58.3**           | **58.3**           | **61.3**           | **64.9**          | **70.0**             | **78.6**             | **84.6**             |
> | 80%          | MCLEA     | 14.6            | 16.5               | 29.2               | 40.9               | 35.8              | 37.4                 | 66.5                 | 71.0                 |
> |              | MEAformer | 29.9            | 38.2               | 51.6               | 52.9               | 36.2              | 43.5                 | 69.9                 | 78.5                 |
> |              | UMAEA     | 24.6            | 26.5               | 31.4               | 43.9               | 35.7              | 43.6                 | 64.3                 | 75.1                 |
> |              | PMF       | 27.7            | 31.7               | 39.5               | 49.6               | 52.4              | 55.2                 | 69.1                 | 78.2                 |
> |              | RULE      | **45.0**        | **55.5**           | **57.2**           | **61.0**           | **62.8**          | **64.0**             | **77.5**             | **83.9**             |

---

> ### Author Response · Authors · 2025-11-17
>
> Table E: Comparisons with state-of-the-art methods on All-attributes setting under extreme noise scenarios.
>
> | Noise Ratios | Method    | ICEWS-WIKI: DNC | ICEWS-WIKI: E-E NC | ICEWS-WIKI: E-A NC | ICEWS-WIKI: A-A NC | DBP15K ZH-EN: DNC | DBP15K ZH-EN: E-E NC | DBP15K ZH-EN: E-A NC | DBP15K ZH-EN: A-A NC |
> | ------------ | --------- | --------------- | ------------------ | ------------------ | ------------------ | ----------------- | -------------------- | -------------------- | -------------------- |
> | 60%          | MCLEA     | 75.9            | 81.7               | 84.9               | 92.0               | 81.0              | 87.4                 | 92.7                 | 92.1                 |
> |              | MEAformer | 91.3            | 92.4               | 94.9               | 95.3               | 92.3              | 92.8                 | 94.8                 | 96.4                 |
> |              | UMAEA     | 85.5            | 86.2               | 88.7               | 91.4               | 90.2              | 91.3                 | 92.6                 | 95.0                 |
> |              | PMF       | 85.7            | 87.1               | 89.7               | 94.8               | 91.1              | 91.4                 | 92.2                 | 96.0                 |
> |              | RULE      | **94.1**        | **95.7**           | **95.4**           | **96.1**           | **94.1**          | **95.1**             | **96.5**             | **96.9**             |
> | 70%          | MCLEA     | 72.2            | 77.8               | 83.5               | 91.4               | 78.1              | 84.9                 | 91.9                 | 91.7                 |
> |              | MEAformer | 88.8            | 91.6               | 94.5               | 95.1               | 90.5              | 91.9                 | 94.4                 | 96.2                 |
> |              | UMAEA     | 83.7            | 84.3               | 87.0               | 91.0               | 88.3              | 90.5                 | 91.8                 | 94.7                 |
> |              | PMF       | 84.7            | 85.5               | 88.7               | 94.1               | 89.3              | 90.4                 | 91.6                 | 95.6                 |
> |              | RULE      | **91.3**        | **95.2**           | **95.2**           | **95.7**           | **92.7**          | **93.9**             | **96.3**             | **96.8**             |
> | 80%          | MCLEA     | 68.1            | 73.6               | 80.9               | 90.4               | 81.4              | 81.8                 | 90.8                 | 91.2                 |
> |              | MEAformer | 87.3            | 90.2               | 94.1               | 94.8               | 88.9              | 89.9                 | 93.5                 | 96.0                 |
> |              | UMAEA     | 75.3            | 81.0               | 85.1               | 89.5               | 85.2              | 88.6                 | 91.0                 | 94.3                 |
> |              | PMF       | 77.3            | 82.2               | 87.3               | 93.1               | 87.2              | 89.0                 | 91.1                 | 95.2                 |
> |              | RULE      | **91.0**        | **94.5**           | **94.9**           | **95.5**           | **90.9**          | **92.8**             | **96.1**             | **96.5**             |
>
>
> The results demonstrate that RULE not only consistently outperforms all the baselines under various noise scenarios, but also exhibits significantly slower performance degradation, which further confirms the robustness of RULE against extreme noise scenarios.

---

> ### Author Response · Authors · 2025-11-17
>
> > Q4: While the TTR module improves the robustness during inference, it may introduce additional computational overhead. The reasoning process involves complex reasoning steps and large language model queries, which could be slow and computationally expensive, especially for large-scale graphs.
>
> **A4**: Thanks for your constructive comments. We acknowledge that employing the test-time reasoning module might introduce additional computational overhead. However, it is worth noting that **we perform MLLM-based reasoning only on the unreliable attribute pairs (Eq. 28) to reduce the computational cost**. In other words, we skip the attribute pairs that satisfy the following conditions:
> $$
> \max(s_{i}^{m}) \ge 0.2
> \quad \lor \quad
> \max(s_{j}^{m}) - s_{ij}^{m} \ge 0.2,
> \qquad
> \forall\, s_{ij}^{m} \neq \max(s_{i}^{m})
> $$
> To further validate the efficiency of the proposed TTR module, we have explored more lightweight designs in Appendix G.8. According to the results in Table 13, we could derive the following observations and conclusions:
>
> i) employing MLLMs such as Qwen2.5-VL 3B and 7B would also lead to a considerable performance boost, **while requiring even less memory cost than the consumption during training time**;
>
> ii) employing the lightweight solutions would **achieve up to a 5× speedup in time cost**;
>
> iii) when resources are limited, RULE can be deployed without MLLMs. **Even without TTR, RULE still significantly outperforms the strongest baselines**, achieving 56.5% vs. 43.9% (best-performing baseline) in the ''Non-name'' setting and 94.0% vs. 91.9% (best-performing baseline) in the ''All-attributes'' setting.
>
> Table 13: Experiment results for the complexity analysis on the ICEWS-WIKI dataset. In the table, the time unit is seconds. "w/o TTR" refers to employing RULE during the training stage, without using the TTR module.
> | Methods        | H@1 (Non-name) | Time Cost (Non-name) | H@1 (All-attributes) | Time Cost (All-attributes) | Memory Consumption  |
> | -------------- | -------------- | -------------------- | -------------------- | -------------------------- | ------------------- |
> | w/o TTR        | 56.5           | 103                  | 94.0                 | 109                        | 1GPU $\times$ ~16GB |
> | Qwen2.5-VL 3B  | 57.2           | 2122                 | 96.5                 | 1008                       | 1GPU $\times$ ~8GB  |
> | Qwen2.5-VL 7B  | 57.4           | 2690                 | 97.1                 | 1329                       | 1GPU $\times$ ~8GB  |
> | Qwen2.5-VL 72B | 58.2           | 10043                | 97.7                 | 4373                       | 8GPU $\times$ ~20GB |

---

> ### Author Response · Authors · 2025-11-17
>
> > Q5: How do you think RULE compares to methods like graph matching or other representation learning-based approaches that also handle noisy correspondences? Some methods like [1] also addresses noise in multi-modal entity alignment, which names semantic consistency. It would be better if you can compare and analyze them.
>
> **A5**: Thanks for your constructive comments. RULE differs from prior noisy correspondence (NC) oriented methods in graph matching or other representation learning domains in the following aspects:
>
> i) **Differences in problem formulation**: most existing noisy correspondence studies tackle the errors in correspondence at a specific level (e.g., pixel-to-pixel or image-to-text),  this work delves into the MMEA task and reveals the specific dual-level noisy correspondence (DNC) problem for the first time;
>
> ii) **Differences in approach design**: existing methods typically adopt a robust objective function to prevent the model from overfitting to incorrect correspondences, while RULE designs robust inter-entity attribute fusion and cross-graph discrepancy elimination to alleviate the negative impact of DNC in MMEA;
>
> iii) **From training-time robustness to test-time robustness**: although existing NC-oriented methods could achieve robustness during the training stage, RULE further enhances robustness at test time, which has yet to be explored in prior work.
>
> As for the paper [D] you mentioned (the paper title was not provided in your description), we infer that it may refer to Towards Semantic Consistency: Dirichlet Energy Driven Robust Multi-Modal Entity Alignment (DESAlign), IEEE ICDE, 2024. We believe the mentioned DESAlign and our RULE differ in the following aspects:
>
> i) **Different Motivations**. DESAlign focuses on addressing disparities in attribute counts or the absence of certain modalities, while RULE aims to mitigate the negative impacts of E-E NC, E-A NC, and A-A NC on intra-entity attribute fusion and inter-graph discrepancy elimination;
>
> ii) **Different Key Ideas.** DESAlign employs a robust semantic learning to mitigate semantic inconsistency and interpolates missing modality through a propagation strategy. In contrast, RULE not only adopts an intra-entity attribute fusion and an inter-graph loss to achieve robustness against DNC, but also leverages the test-time reasoning module to mine underlying associations between attributes and thus guarantees more accurate equivalent entity identification.
>
> **In the revised manuscript, we have cited the related work DESAlign and supplemented the detailed discussions about the differences between DESAlign and our work in Appendix G.14.**
>
> [A] MEAformer: Multi-modal Entity Alignment Transformer for Meta Modality Hybrid, MM, 2023.
>
> [B] Progressively Modality Freezing for Multi-Modal Entity Alignment, ACL, 2024.
>
> [C] Rethinking Uncertainly Missing and Ambiguous Visual Modality in Multi-Modal Entity Alignment, ISWC, 2023.
>
> [D] Towards Semantic Consistency: Dirichlet Energy Driven Robust Multi-Modal Entity Alignment, IEEE ICDE, 2024.

---

### Official Review · Reviewer_65Cw · 2025-11-23

**Soundness:** 3
**Presentation:** 4
**Contribution:** 3
**Rating:** 8
**Confidence:** 4

**Summary:**

This paper addresses the overlooked problem of Dual-level Noisy Correspondence in multi-modal entity alignment, where both intra-entity attributes and inter-graph alignment pairs contain substantial noise. The authors propose RULE, a unified framework that estimates pair reliability through uncertainty and consensus modeling, performs noise-aware robust learning and multi-modal fusion during training, and further integrates a test-time reasoning module using an MLLM to refine alignment decisions. Experiments on several benchmarks demonstrate that RULE significantly improves robustness across a wide range of noise levels and achieves clear gains over state-of-the-art methods.

**Strengths:**

S1. The paper convincingly argues that Dual-level Noisy Correspondence is ubiquitous in real MMKG scenarios, yet largely ignored by prior MMEA works. The motivation is well-founded and supported by quantitative evidence of high noise ratios.

S2. RULE provides an end-to-end framework that integrates reliability estimation, noise-aware learning, multi-modal fusion, and test-time reasoning. The combination of training-time robustness and inference-time semantic reasoning is novel and compelling.

S3. Experiments cover multiple datasets, modalities, and a wide spectrum of noise levels (0%–70%). Results show large improvements over SOTA baselines in both low-noise and extremely noisy regimes, demonstrating the practical robustness of RULE.

S4. The evidential uncertainty modeling and greedy consensus estimation are well-justified. Visualizations (e.g., Fig. 3) clearly show distinguishable distributions between clean and noisy correspondences.

**Weaknesses:**

W1. RULE contains several components (uncertainty estimation, consensus reasoning, robust learning, fusion, test-time MLLM reasoning). While each part is motivated, the overall system introduces significant complexity, making it hard to isolate which component contributes most.

W2. Although TTR appears effective, the paper lacks a thorough analysis of computational overhead, scalability, and failure modes. Using a large MLLM at inference time raises concerns for deployment.

W3. Although TTR is conceptually strong, the paper could benefit from more real examples illustrating how MLLM reasoning corrects embedding-based mistakes, especially under ambiguous or incomplete attribute sets.

**Questions:**

1. How sensitive is RULE to the thresholds in pair division? Are there universal settings across datasets, or does each dataset require tuning?

2. What is the computational overhead of TTR? How many calls to the MLLM are required per entity, and is inference feasible for large-scale KG alignment?

---

> ### Author Response · Authors · 2025-11-25
>
> Thanks for the insightful reviews. We will answer your questions one by one in the following.
>
> > Q1: RULE contains several components (uncertainty estimation, consensus reasoning, robust learning, fusion, test-time MLLM reasoning). While each part is motivated, the overall system introduces significant complexity, making it hard to isolate which component contributes most.
>
> **A1**: Thank you for your comments. Actually, the key idea of RULE is simple and straightforward. In brief, we aim to tackle the dual noisy correspondence (DNC) problem by first estimating the reliability of each correspondence type and then leveraging it to achieve training-time robustness and test-time robustness.
>
> Accordingly, we address your concerns regarding which component contributes most from three perspectives: **reliability estimation**, **training-time robustness**, and **test-time robustness**. The corresponding ablation studies are presented in Table 3, where one could observe that
>
> - Regarding the reliability estimation (Line ③ and ④), the uncertainty principle (Sec 2.1.1) contributes more in the reliability estimation, and combining it with the consistency principle (Sec 2.1.2) would further improve the performance.
> - Regarding the training-time robustness (Line ① and ②), the robust learning (Sec 2.3) contributes more, as it could eliminate the negative impact of inter-graph NC during the cross-graph discrepancy elimination process.
> - Regarding the test-time robustness (Line ⑤, ⑥, and ⑦), the robust fusion (Sec 2.4) contributes the most in this stage, as it could effectively integrate both the prior results and the MLLM reasoning results.
>
> As a result, we conclude that **each component acts as an inseparable role for achieving DNC-robust MMEA**.
>
> Table 3: Ablation study of the various modules in the train and test stages.
>
> | Stage | Setting      | Non-name H@1 | Non-name H@5 | Non-name MRR | All-attributes H@1 | All-attributes H@5 | All-attributes MRR |
> | ----- | ------------ | ------------ | ------------ | ------------ | ------------------ | ------------------ | ------------------ |
> | Train | w/o DRL      | 31.6         | 45.9         | 38.6         | 82.3               | 90.4               | 86.0               |
> |       | w/o DRF      | 50.4         | 66.2         | 57.6         | 93.4               | 97.4               | 95.2               |
> |       | Only Unc.    | 53.5         | 67.8         | 60.2         | 93.6               | 97.4               | 95.4               |
> |       | Only Cons.   | 48.3         | 60.3         | 54.3         | 87.7               | 93.2               | 90.4               |
> | Test  | w/o DRF      | 52.4         | 66.2         | 59.0         | 95.1               | 97.9               | 96.3               |
> |       | w/o TTR      | 56.5         | 68.0         | 62.3         | 94.0               | 97.7               | 95.7               |
> |       | MLLM Enhance | 56.6         | 69.0         | 62.4         | 97.6               | 98.2               | 97.9               |
> | Both  | Default      | 58.2         | 69.7         | 63.6         | 97.7               | 98.3               | 98.0               |

---

> > ### Author Response · Authors · 2025-11-25
> >
> > > Q2: Although TTR appears effective, the paper lacks a thorough analysis of computational overhead, scalability, and failure modes. Using a large MLLM at inference time raises concerns for deployment. (Weakness 2)
> > >
> > > How many calls to the MLLM are required per entity, and is inference feasible for large-scale KG alignment? (Question 2)
> >
> > **A2**: Thank you for your valuable comments. We will address your concerns one by one in the following.
> >
> > + **Computational Overhead Analysis**
> >
> > We acknowledge that employing the test-time reasoning module might increase the computational overhead. In the work, we have tried to reduce the computational cost from two perspectives.
> >
> > First, **we perform MLLM-based reasoning only on the unreliable attribute pairs (Eq. 28), thus reducing the computational overhead to a great extent**. To be specific, we skip the attribute pairs that satisfy the following conditions:
> > $$
> > \max(s_{i}^{m}) \ge 0.2
> > \quad \lor \quad
> > \max(s_{j}^{m}) - s_{ij}^{m} \ge 0.2,
> > \qquad
> > \forall\, s_{ij}^{m} \neq \max(s_{i}^{m})
> > $$
> > Second, **we have explored various MLLM choices and given an analysis of the corresponding computational overhead and scalability in Table 13 (Appendix G.8)**. According to the results, one could derive the following observations and conclusions:
> >
> > i) **employing smaller-size MLLMs such as Qwen2.5-VL 3B and 7B would significantly reduce the computational cost (5x) and memory cost (20x), while keeping very competitive performance**.
> >
> > iii) when resources are limited, RULE can be deployed without MLLMs (i.e., w/o TTR in Table 13). Note that, **even without TTR, RULE still significantly outperforms the strongest baselines**, achieving 56.5% vs. 43.9% (best-performing baseline) in the ''Non-name'' setting and 94.0% vs. 91.9% (best-performing baseline) in the ''All-attributes'' setting.
> >
> > Table 13: Experiment results for the complexity analysis on the ICEWS-WIKI dataset. The time unit is seconds. "w/o TTR" refers to employing RULE during the training stage, without using the TTR.
> >
> > | Methods        | H@1 (Non-name) | Time Cost (Non-name) | H@1 (All-attributes) | Time Cost (All-attributes) | Memory Consumption  |
> > | -------------- | -------------- | -------------------- | -------------------- | -------------------------- | ------------------- |
> > | w/o TTR        | 56.5           | 103                  | 94.0                 | 109                        | 1GPU $\times$ ~16GB |
> > | Qwen2.5-VL 3B  | 57.2           | 2122                 | 96.5                 | 1008                       | 1GPU $\times$ ~8GB  |
> > | Qwen2.5-VL 7B  | 57.4           | 2690                 | 97.1                 | 1329                       | 1GPU $\times$ ~8GB  |
> > | Qwen2.5-VL 72B | 58.2           | 10043                | 97.7                 | 4373                       | 8GPU $\times$ ~20GB |
> >
> > + **Scalability Analysis**
> >
> > To the best of our knowledge, the DBP15K benchmark used in our paper might be one of the **largest** MMEA benchmarks, while the ICEWS benchmark might contain the **most complex** relationship structures. Therefore, **we have evaluated the proposed RULE on the two challenging benchmarks to verify the inference feasibility on large-scale KG alignment**.
> >
> > To further explore the inference feasibility, we consider a potential setting where the test-time reasoning module is applied to an unprecedented large-scale MMKG with more than 1M entities. As shown in Table 13, lightweight MLLM solutions (e.g., Qwen2.5-VL 3B/7B) achieve a test-time reasoning speed of approximately 1.0 seconds per entity. In this case, **an 8×3090 GPU setup would require roughly 36 hours to finish test-time reasoning on the MMKG with 1M entities**, which further supports the practical inference feasibility of the TTR module on large-scale scenarios. **We believe that this cost is acceptable in the current era of high-performance large-model computation.**
> >
> > + **MLLM Calls**
> >
> > For one specific attribute, a given query entity requires **at most 10 MLLM calls**. It is worth noting that we perform MLLM-based reasoning only on the unreliable attribute pairs (Eq. 28), which significantly reduces the number of MLLM calls and the corresponding computational cost.

---

> > > ### Author Response · Authors · 2025-11-25
> > >
> > > + **Failure Cases**
> > >
> > > In response to your concerns，**we conduct ADDITIONAL visualization analysis about the failure cases of the TTR module in Fig. 13**. According to the visualization results, we attribute these failure cases to the following two reasons.
> > > On the one hand, **the MLLM might lack enough knowledge to perform correct reasoning**. For example, when the query refers to the abbreviation of Toronto Metropolitan University, the MLLM does not possess the necessary background knowledge and thus cannot find the corresponding university logo.
> > > On the other hand, **the MLLM might sometimes resort to shortcut cues rather than performing deep reasoning**, leading to undiscovered associations even after reasoning. For instance, the MLLM may still rely on visual similarity when identifying coastal cities, while ignoring underlying semantic cues that indicate the corresponding country.
> > >
> > > **In the revised manuscript, we have included the above detailed discussions of the failure cases in Appendix I.**
> > >
> > > > Q3: Although TTR is conceptually strong, the paper could benefit from more real examples illustrating how MLLM reasoning corrects embedding-based mistakes, especially under ambiguous or incomplete attribute sets.
> > >
> > > **A3**: Thanks for your valuable suggestions. In the following, we will address your concerns one by one.
> > >
> > > + **Embedding-based Mistakes**
> > >
> > > In the original submission, we have provided several visualization examples in Figs. 11 and 12 (Appendix I). According to these results, we observe that the TTR module could **mine the underlying associations between attributes, thus addressing the embedding-based mistakes**. For instance, different textual expressions of the same entity (e.g., Brick City and Newark, New Jersey), different images of the same entity (e.g., the Qatar national flag and a map of Qatar). In other words, **the MLLM could uncover seemingly dissimilar but inherently identical attributes, which is often overlooked by existing MMEA methods**.
> > >
> > > + **Ambiguous Attributes**
> > >
> > > In response to your concerns, **we conduct ADDITIONAL visualization analysis about the ambiguous attributes in Fig. 14**. From the results, TTR could perform correct test-time reasoning even when the attributes are ambiguous. For example, the image attributes of the candidates are highly similar, as they all depict visually similar individuals. Nevertheless, the TTR module is able to uncover underlying associations, such as identifying the same individual across different scenes or recognizing variations in clothing and appearance.
> > >
> > > + **Incomplete Attributes**
> > >
> > > **To address your concerns, we carry out ADDITIONAL visualization analysis about the missing attributes in Fig. 15**. From the results,  we observe that when the image attribute of the correct candidate is missing, the TTR module assigns low similarity scores to other images and instead relies on the available and more reliable textual attributes for more accurate reasoning. As a result, the TTR module could still achieve correct alignment even in the presence of missing attributes, demonstrating the effectiveness and robustness of RULE.
> > >
> > > **In the revised manuscript, we will include detailed discussions about the ambiguous attributes and incomplete attributes in Appendix I.**

---

> ### Author Response · Authors · 2025-11-25
>
> > Q4: How sensitive is RULE to the thresholds in pair division? Are there universal settings across datasets, or does each dataset require tuning?
>
> **A4**: Thank you for your comments. In the implementation, we adopt a unified threshold of $\beta = 0.3$ across all datasets and settings, which already yields substantial performance gains. To further address your concern, **we conduct ADDITIONAL experiments to evaluate the sensitivity of RULE with respect to the threshold $\beta$**. Specifically, we investigate the influence of $\beta$ under injected DNC ratios of 0%, 20%, 50%, and 70% on the ICEWS-WIKI benchmark. The corresponding results are summarized in the following tables.
>
> Table A: Hyperparameters Analysis of the Threshold $\beta$ under various noise settings regarding the Hits@1 metric.
>
> | Threshold $\beta$ | Non-name (Inherent DNC) | Non-name (20% DNC) | Non-name (50% DNC) | Non-name (70% DNC) | All-attributes (Inherent DNC) | All-attributes (20% DNC) | All-attributes (50% DNC) | All-attributes (70% DNC) |
> | ---------------------- | ----------------------- | ------------------ | ------------------ | ------------------ | ----------------------------- | ------------------------ | ------------------------ | ------------------------ |
> | 0                      | 59.3                    | 56.8               | 55.7               | 38.9               | 94.5                          | 93.7                     | 93.3                     | 89.6                     |
> | 0.1                    | 60.1                    | 58.1               | 55.9               | 41.2               | 95.7                          | 94.6                     | 93.4                     | 90.2                     |
> | 0.2                    | 61.2                    | 58.4               | 56.2               | 46.5               | 96.3                          | 95.4                     | 93.5                     | 91.0                     |
> | 0.3                    | **62.1**                | **60.4**           | **56.5**           | **48.2**           | **96.7**                      | **95.8**                 | **94.0**                 | **91.3**                 |
> | 0.4                    | 61.4                    | 59.1               | 56.1               | 47.3               | 96.4                          | 95.7                     | 93.6                     | 90.7                     |
> | 0.5                    | 59.6                    | 57.3               | 55.8               | 45.3               | 96.1                          | 94.9                     | 93.6                     | 90.3                     |
> | 0.6                    | 57.4                    | 54.2               | 53.5               | 41.6               | 94.6                          | 93.8                     | 93.4                     | 89.6                     |
> | 0.7                    | 48.9                    | 50.2               | 52.8               | 35.8               | 93.5                          | 92.7                     | 92.3                     | 88.9                     |
>
> From the results, **one could observe that RULE exhibits stable performance when $\beta$ within $[0.2, 0.4]$ and achieves the best performance with threshold $\beta = 0.3$**.

---

### Author Response · Authors · 2025-12-03
**Summary of Revision**

Dear AC and SAC,

We sincerely thank you for your time and effort in evaluating our paper. Below, we provide a brief summary of the rebuttal process to assist your assessment.

We are encouraged by the **positive scores (Rating 8, 6, 8, 8)** and comments from all four reviewers. All reviewers (Reviewer 65Cw, Xav9, ny9k, 1ndq) reached a consensus recognizing that the paper addresses **a practical and new problem** in the multi-modal entity alignment task, and the proposed method is **technically sound and novel**. We are also glad that they find our extensive evaluation results under various noise setups are thorough and convincing (Reviewer 65Cw, ny9k, 1ndq).

Beyond these positive comments, we carefully addressed all concerns raised by the reviewers. Our responses are summarized below:

- We provide an in-depth analysis about the computational overhead of the test-time reasoning module and include the corresponding results in Appendix G.8. (Reviewers 65Cw, Xav9, ny9k)
- We conduct additional parameter analyses under more noise settings to comprehensively investigate the influence of three parameters as requested. (Reviewer 65Cw, Xav9)
- We provide additional discussions with related works in Appendix G.14 (Reviewer 65Cw, 1ndq), reliability estimation strategy in Appendix G.15 (Reviewer ny9k), and model-agnostic nature of our method in Appendix G.11 (Reviewer ny9k).
- We carry out additional visualization analyses about the working mechanism and failure modes of our test-time reasoning module, and include these results in the Appendix I. (Reviewer 65Cw, 1ndq)
- We conduct additional evaluations to further verify the robustness of the proposed method under extreme noise scenarios. (Reviewer Xav9)
- We provide more analyses and explanations regarding the importance of different components (Reviewer 65Cw), network details (Reviewer Xav9), and the noise setup (Reviewer 1ndq).

In addition, we would like to take this opportunity to discuss and summarize the potential insights that this work brings to the community:

- **Encouraging future studies beyond embedding-based similarity**: Existing approaches typically establish correspondence between two samples by computing the similarity between their embeddings. However, such embedding-based paradigm only capture the surface-level similarity, while largely overlooking the potential connections among samples. For example, images of the football player ''Cristiano Ronaldo'' and his home country ''Portugal'' may appear visually disparate, yet they are inherently associated. Notably, such underlying correspondences are pervasive across nearly all multi-modal tasks. Accordingly, it is desirable to develop effective mechanisms (e.g., test-time reasoning module in our work) to prevent seemingly dissimilar but inherently identical samples from being neglected.
- **Facilitating MMEA on large-scale datasets**: Although commonly used MMEA datasets are small in scale (i.e., containing up to 30K entity pairs) and carefully annotated by humans, they still exhibit an average noise rate of over 50% (Appendix B). Such a DNC challenge would become even more severe in large-scale knowledge graphs such as Wikidata, DBpedia, and YAGO datasets (i.e., containing over 10M–100M entities). We believe that DNC challenge is one of the key reasons why large-scale datasets are currently lacking in the MMEA field. The proposed RULE framework offers a promising solution toward achieving entity alignment on larger-scale datasets with the DNC challenge.

We reply to each reviewer's questions in detail below their reviews. Please kindly check them out. Thank you and please feel free to ask any further questions.

Best Regards,

Authors of Paper 12148

---

### Meta-Review · Area_Chair_zUGe · 2026-01-11

**Summary:**

The paper identifies and formalizes a practical yet previously overlooked problem in multi-modal entity alignment (MMEA): Dual-level Noisy Correspondence (DNC)—noise in both intra-entity (entity–attribute) and inter-graph (entity–entity, attribute–attribute) alignments. The authors propose RULE, a robust framework that estimates correspondence reliability via uncertainty and consensus, performs noise-aware training, and introduces a test-time reasoning (TTR) module using an MLLM to uncover hidden semantic associations. All four reviewers recognize the strong motivation, novel problem formulation, and comprehensive empirical validation across multiple benchmarks and noise settings. Concerns mainly center on computational overhead of TTR, model complexity, and need for deeper analysis of failure modes—all of which were thoroughly addressed in the rebuttal with new experiments, ablation studies, and efficiency analyses.

**Reviewer Concerns:**

Addressed by rebuttal:

Computational cost of TTR: Authors showed TTR is applied only to unreliable pairs, reducing calls; provided runtime/memory comparisons across MLLM sizes (3B/7B/72B); demonstrated that even without TTR, RULE outperforms SOTA.
Hyperparameter sensitivity: Added extensive ablation under 0–80% noise for λ, β, τ, showing stable performance with default values (e.g., β=0.3).
Extreme noise robustness: Conducted new experiments at 60–80% noise across DNC variants (E–E, E–A, A–A NC), confirming consistent gains over baselines.
Model generality: Verified effectiveness with alternative backbones (SigLIP, BLIP), proving model-agnostic nature.
Interpretability of TTR: Provided visualizations of MLLM correcting embedding-based errors (e.g., “Brick City” ↔ “Newark”), handling ambiguous/missing attributes, and analyzing failure cases (e.g., knowledge gaps, shortcut reliance).
Relation to prior NC methods: Clarified that existing approaches handle single-level noise and lack test-time reasoning; RULE is the first to address dual-level noise with full-process robustness.
Still outstanding (minor):

Deployment feasibility of MLLM-based TTR: While lightweight options (3B/7B) reduce cost, real-world large-scale deployment may still face latency/budget constraints—though authors note TTR is optional.
Theoretical justification for consensus principle: Empirical evidence is strong, but a formal proof linking consensus to ground-truth recovery remains limited.

Overall, all major concerns were substantively resolved.

**Reviewer Scores:**

65Cw (initial: 8 – accept): Already highly positive; rebuttal strengthened confidence with ablations and overhead analysis. Would maintain 8.
Xav9 (initial: 6 – marginally above threshold): Raised valid concerns on hyperparameters and extreme noise; authors provided comprehensive new results. Likely upgrades to 7 or 8.
ny9k (initial: 8 – accept): Appreciated clarity and experiments; minor questions on generalization and complexity fully answered. Would maintain 8.
1ndq (initial: 8 – accept): Praised problem novelty and evaluation; requested visualizations and comparisons—both delivered. Would maintain 8.

---

### Decision · Program_Chairs · 2026-01-26

Accept (Oral)